# Phenotypically complex living materials containing engineered cyanobacteria

Debika Datta [1,5], Elliot L. Weiss [2,3,5], Daniel Wangpraseurt [1,2], Erica Hild[1], Shaochen Chen [1], James W. Golden [3], Susan S. Golden [3] ✉ & Jonathan K. Pokorski [1,4] ✉

The field of engineered living materials lies at the intersection of materials science and synthetic biology with the aim of developing materials that can sense and respond to the environment. In this study, we use 3D printing to fabricate a cyanobacterial biocomposite material capable of producing multiple functional outputs in response to an external chemical stimulus and demonstrate the advantages of utilizing additive manufacturing techniques in controlling the shape of the fabricated photosynthetic material. As an initial proof-of-concept, a synthetic riboswitch is used to regulate the expression of a yellow fluorescent protein reporter in *Synechococcus elongatus* PCC 7942 within a hydrogel matrix. Subsequently, a strain of *S. elongatus* is engineered to produce an oxidative laccase enzyme; when printed within a hydrogel matrix the responsive biomaterial can decolorize a common textile dye pollutant, indigo carmine, potentially serving as a tool in environmental bioremediation. Finally, cells are engineered for inducible cell death to eliminate their presence once their activity is no longer required, which is an important function for biocontainment and minimizing environmental impact. By integrating genetically engineered stimuli-responsive cyanobacteria in volumetric 3D-printed designs, we demonstrate programmable photosynthetic biocomposite materials capable of producing functional outputs including, but not limited to, bioremediation.

Synthetic stimuli-responsive polymeric materials have been fabricated to sense and respond to different environmental conditions such as chemicals, pH, light, and temperature. Such polymers have been utilized for applications including therapeutics[1], drug delivery[2], biomedical devices[3], biosensors[4,5], electronics[6,7], and soft robotics[8]. However, in most cases, the ability of these materials to respond with high specificity and produce the desired output is limited by the nature of the available stimuli and is restricted by the specifics of a chemically driven response. The development of purely synthetic materials is therefore limited by the coupling of available external cues as well as the extent and nature of the material's response.

Unlike traditional materials systems, biological systems innately respond and adapt to their surrounding environment and can generate bioproducts with a wide range of functions. The field of synthetic biology aims to exploit these natural systems by engineering them to perform high-value user-defined functions for the production of biofuels, proteins, fertilizers, bioplastics, and catalysts[9–13]. Recent progress in synthetic biology enables the design and manipulation of cellular signaling pathways for regulating gene expression for the

[1]Department of Nanoengineering, University of California San Diego, La Jolla, CA, USA. [2]Scripps Institution of Oceanography, University of California San Diego, La Jolla, CA, USA. [3]Department of Molecular Biology, University of California San Diego, La Jolla, CA, USA. [4]Center for Nano-ImmunoEngineering and Institute for Materials Discovery and Design, University of California San Diego, La Jolla, CA, USA. [5]These authors contributed equally: Debika Datta, Elliot L. Weiss. ✉e-mail: sgolden@ucsd.edu; jpokorski@ucsd.edu

constitutive or induced (in response to an environmental stimulus) production of a desired biosynthetic output.

The emerging field of engineered living materials (ELMs) aims to design programmable materials[14–21]. Compared to synthetic stimuli-responsive materials, ELMs use genetically engineered biological components integrated into a composite material to produce functional outputs in response to environmental cues. Recent ELMs have utilized a diversity of microorganisms including bacteria, yeast, fungi, and algae. Some notable examples include the fabrication of skin patches for wound healing[22], sweat-responsive biohybrid fabrics[23], a biodegradable aqua plastic[24], the incorporation of chloroplasts for self-repairing hydrogels[25], and photosynthetic $O_2$ generators for enhancing mammalian cell viability[26]. While the potential of ELMs is vast, there are significant challenges in their development including the necessity for genetically stable engineered strains, the formulation of materials to support the growth and viability of engineered cells over long durations of time, and programming of cells to be stimuli-responsive at the material interface. Thus, an appropriate pairing of biological components and polymeric materials is critical to developing any stimuli-responsive ELM.

With the aim of developing innovative sustainable materials and overcoming the limitations of traditional stimuli-responsive materials, we incorporate photosynthetic cyanobacterial cells within a polymer composite material. Cyanobacteria are appealing candidates for inclusion in ELMs due to their viability in low-cost media, with the added benefit that the carbon source for these phototrophs, $CO_2$, is freely available. Moreover, the abundance of genetic engineering tools available for several cyanobacterial species allows for modular systems of plasmid design and gene expression controlled by external cues. Engineered plasmids can be self-replicating or designed for the integration of heterologous genes into the chromosome.

Over the last decade, a number of tools have been developed and used for genetic modification in cyanobacteria and applied for heterologous expression of high-value products and regulatory circuits[27–29]. Notably, the development of various cyanobacteria-compatible promoters, ribosomal binding sites (RBSs), reporters, modular vectors, and markerless selection systems have contributed greatly to the evolution of cyanobacterial synthetic biology[30,31]. Of relevance to this study is the use of a riboswitch as a tool for regulating gene expression. Riboswitches are aptameric sequences in the mRNA that regulate gene expression in response to ligand binding and can be used to alter rates of protein production. Previous studies have demonstrated the utility of the theophylline-responsive riboswitch-F in the cyanobacterium *Synechococcus elongatus* PCC 7942 (*S. elongatus*), which, when supplemented with the small molecule theophylline, undergoes a conformational change exposing the RBS and allows the mRNA to bind to a ribosomal subunit, resulting in the translation of a target protein[32]. In the stimuli-responsive material presented in this study, the translation of proteins from heterologous genes of interest is regulated in the cyanobacteria-polymer composite using a chemically responsive synthetic riboswitch. Bioremediation is an area where ELMs containing genetically modified organisms are likely to provide substantial benefit[33,34]. Previous work has demonstrated that cyanobacteria are capable of producing both native and recombinant enzymes with relevance to bioremediation[35–37].

In this work, we aim to develop stimuli-responsive ELMs that decontaminate chemical pollutants by regulating the production of a laccase enzyme capable of degradation. Furthermore, we engineer cyanobacteria to have a 'kill switch' and show the induction of lytic cell death upon riboswitch activation, limiting biofouling in the case of cells leaching from the ELM. The ELM presented in this study responds to chemical stimuli with phenotypically complex outputs beyond that which could be achieved in a fully synthetic system.

## Results

### Fabrication of ELMs with *S. elongatus* and alginate hydrogels

Several ELMs were fabricated that contain engineered strains of *S. elongatus* expressing heterologous proteins, and the material properties, cell viability, and biological activity of the composite materials were investigated. ELMs were constructed to contain *S. elongatus* strains engineered to express a variety of proteins under the regulation of a theophylline-responsive riboswitch to demonstrate stimuli-responsive photosynthetic biomaterials. Figure 1 illustrates a schematic overview of the fabrication of these ELMs that combines genetically engineered cyanobacterial cells with an alginate polymer to create a 3D-printed hydrogel composite material; the gene of interest in the cells is activated by the presence of an inducer molecule (input) producing a specific protein of interest (output).

### 3D printing of *S. elongatus* cells

To create complex geometric cellular scaffolds, we fabricated cell-laden polymeric gels using direct-ink-writing (DIW) additive manufacturing techniques. Along with the transparency and porosity of the hydrogel material, another critical advantage of using a gel-based ink is the ability to protect the cells from mechanical shear stress during the extrusion process. This strategy minimizes damage to cell membranes, enhancing cell viability in the printed pattern. Identifying a biocompatible and printable polymer is critical when encapsulating microorganisms in any 3D-printed matrix. Initial optimization experiments were conducted to identify a suitable material for encapsulating cyanobacterial cells that would sustain the growth and viability of the cells at ambient temperatures and light levels. Several natural and synthetic polymers capable of forming hydrogels were tested and our chosen candidate was an alginate hydrogel. The transparency of the hydrogel allows for light penetration, while the presence of relatively large pores within the matrix facilitates diffusion of gas and nutrients[38]; these properties and a capacity for water retention make alginate an excellent candidate for photosynthetic ELMs. Alginate is a natural polysaccharide from seaweed composed of β-D-mannuronic acid and α-L-guluronic acid that is physically cross-linked using divalent cations[39]. The optimal condition for alginate to be cross-linked and printable at room temperature was determined to be a mixture of 4% w/v alginate solution with a 50 mM $CaSO_4$ slurry suspension in a 2:1 ratio, resulting in a final 2.67% w/v alginate solution in 16.67 mM $CaSO_4$. A crucial step in DIW printing is obtaining a material with suitable viscosity such that the printed structure can retain its shape post-printing[40]. This property was achieved by adding a solution of 100 mM $CaCl_2$ to the hydrogel scaffold post-printing for 15 min to strengthen and stabilize the gel structure. The ink formulation was stable enough to retain its shape for various patterns, consisting of 5–17 layers of bioink (Supplementary Data 1). For hydrogels embedded with cyanobacterial cells, the printed patterns were incubated under light in BG-11 medium post-printing to promote growth. To achieve an optimal gel formulation allowing for both 3D bioprinting and viability of the biological component, we utilized the same two-stage crosslinking strategy to prepare our gel-based bioink, producing a pattern with high structural integrity.

### Mechanical properties of hydrogels

Rheological measurements were carried out to test the mechanical properties of the 3D-printed hydrogel with and without wild-type (WT) *S. elongatus* cells. The viscoelastic moduli with increasing oscillatory strain in both the unloaded gel (printed alginate hydrogel without cells) and the bioink (cell-laden gel) exhibited similar storage modulus values at lower oscillatory strain (Supplementary Fig. 1a). Additionally, we observed a shift in crossover frequencies between the unloaded gel and the bioink samples (Supplementary Fig. 1a); the viscosity of the bioink was higher than the virgin alginate inks (Supplementary Fig. 1b). The gels demonstrated a shear-thinning behavior, a key requirement

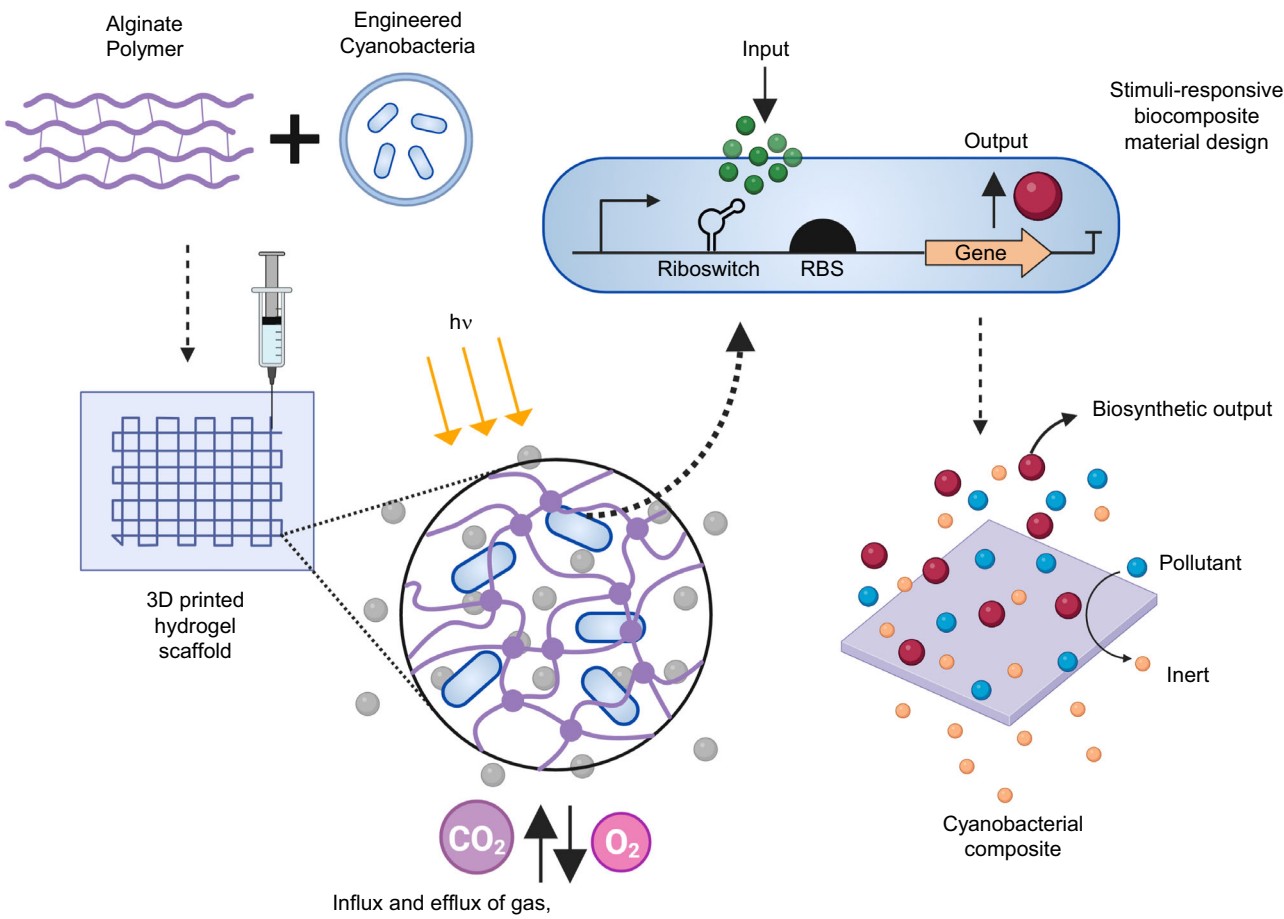

**Fig. 1 | Schematic illustration of the use of engineered cyanobacteria for creating stimuli-responsive living materials used in this study.** Genetically engineered strains of the obligate photoautotroph *S. elongatus* were immobilized in a matrix of the natural polymer alginate. The chosen polymer scaffold can be swollen in aqueous conditions, allows for the transport of nutrients and gases within the matrix, and provides a suitable microenvironment for embedded cyanobacterial growth and the induction of gene expression by external cues.

for DIW-based printing. While highly viscous inks are favorable for increasing print resolution, too high of viscosity can result in clogging of the print needle and reduced cell viability. In this study, needle clogging was observed when the alginate concentrations were greater than ~3% w/v, highlighting the necessity of a balance between printability, hydrogel integrity, and cell growth and viability. Furthermore, storage modulus (G′), loss modulus (G″), and viscosity are influenced by factors including alginate concentration, molecular weight, crosslinker concentration (at both stages of printing), degree of crosslinking, and concentration of cells. Although we were unable to quantify the precise number of cells within the hydrogel matrix at any given time point due to experimental limitations, we observed that the presence of cells increases the storage modulus of the hydrogel between an oscillation strain range of ~1-10%.

**Growth and viability of *S. elongatus* cells in alginate hydrogels**
Over seven days of growth, the biomass of WT *S. elongatus* cells within various designs of hydrogel constructs became visibly denser (as indicated by dark green coloration); however, designs that had low surface area to volume ratios, such as a disk design (2 cm in diameter), resulted in a decrease in biomass towards the center of the hydrogel, suggesting a limitation of gas and/or nutrient exchange (Fig. 2a). For this reason, subsequent experiments were conducted using hydrogels printed in a 29 × 29 mm grid configuration to achieve a higher surface area to volume ratio to enhance mass transport. While an enhanced surface area to volume ratio was targeted in the study, the thickness, as

well as density of printed layers, will also likely affect the growth behavior of cyanobacterial ELMs. The grid-patterned hydrogel resulted in a greater density of biomass per unit volume hydrogel, highlighting the utility of additive manufacturing to impact geometry, and hence, the viability of the ELMs. Field emission scanning electron microscopy (FESEM) images of an unloaded gel illustrate the porous nature of the hydrogel scaffold ((Fig. 2b (i & ii) and Supplementary Fig. 2a, b), with apparent pore sizes ranging from less than 40 μm to over 60 μm. Micrographs of hydrogels printed with bioink illustrate a combination of individual cells and colonies of cells adhered to the hydrogel surface ((Fig. 2b (iii & iv) and Supplementary Fig. 2c–e). Enlarged FESEM images are provided in Supplementary Fig. 2. Given an average *S. elongatus* cell size of approximately 2 μm, and that the pore sizes observed in the biocomposite hydrogel often exceeded 60 μm, the pore diameter is likely not a limitation to the growth of cyanobacteria within the matrix. To visualize the proportion of viable to dead cells within the bioprinted matrix, the resultant hydrogels were imaged using confocal microscopy for chlorophyll autofluorescence and cell death using SYTOX Blue, a blue nucleic acid stain that penetrates cells with damaged cell membranes (Fig. 2c and Supplementary Figs. 3–5). Red fluorescent patches, indicative of healthy colonies of cyanobacteria, were visible throughout the hydrogel over seven days of growth, whereas blue patches indicative of dead cells were distributed heterogeneously in the hydrogel. Additional brightfield and superimposed images from confocal data of a hydrogel containing WT *S. elongatus* cells from 0 days and 7 days post-printing are available in

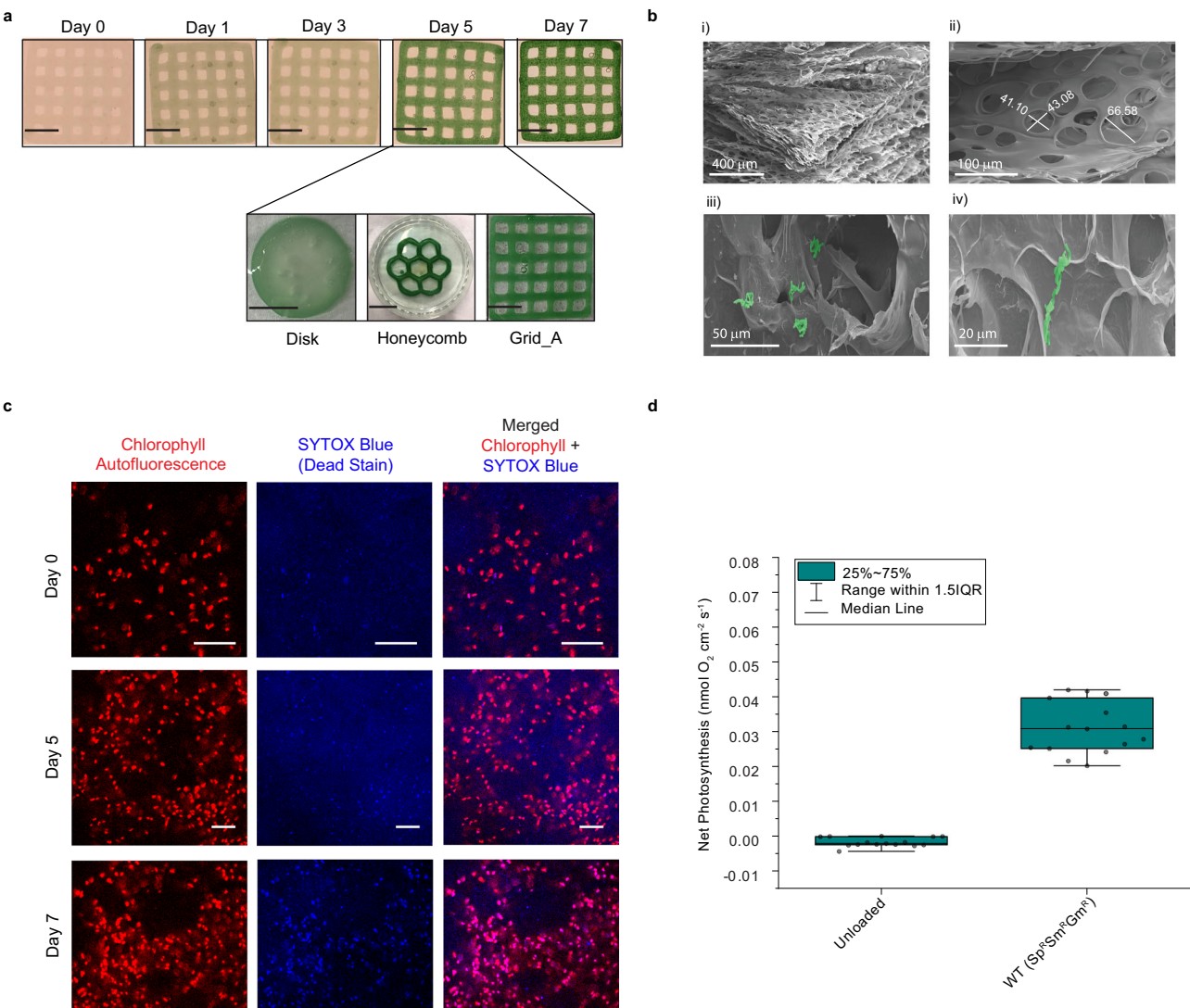

**Fig. 2 | Cell viability within hydrogels. a** Top: growth of WT *S. elongatus* within a 3D-printed grid pattern over 7 days. Bottom: representative images of 5-day-old hydrogels containing WT cells printed in disk, honeycomb, and grid_A geometries. The dimension details of these different patterns are described in Supplementary Data 1. Scale bars in lower left correspond to 10 mm. Experiments were conducted thrice with independent samples ($n = 3$). **b** Representative FESEM images of unloaded hydrogels (i & ii), and hydrogels containing WT *S. elongatus* cells (iii & iv). *S. elongatus* cells are highlighted in false color green. Experiments were conducted with independent samples ($n = 3$). **c** Representative confocal microscopy images of chlorophyll autofluorescence and SYTOX blue staining of a hydrogel containing

WT *S. elongatus* cells after 0, 5, and 7 days of growth post-printing. Experiments were conducted with independent samples ($n = 3$ for Days 0 and 7). Scale bar corresponds to 50 μm. **d** Box plots with center lines showing the medians, boxes indicating the lower (25%) and upper (75%) quartiles, and whiskers indicating 1.5× interquartile range of net photosynthesis at an incident downwelling irradiance of 80 μmol photon m$^{-2}$ s$^{-1}$ for an unloaded hydrogel and an antibiotic-resistant *S. elongatus* strain [WT(Sp$^R$Sm$^R$Gm$^R$)] encapsulated in the hydrogel. Experiments were performed as measurements of independent regions of a hydrogel ($n = 15$). Error bars are S.E.M.

Supplementary Fig. 5. An assessment of cell viability from confocal image analyses of a hydrogel containing WT *S. elongatus* shows 94.9 ± 2.2% and 76.8 ± 8.2% of total cells to be viable on Day 0 and Day 7 after printing, respectively (Supplementary Fig. 6). A representative 3D reconstruction of confocal z-stacked imaging data for the hydrogel containing WT *S. elongatus* cells shows the presence of both live and dead cells in a 3D space (Supplementary Fig. 7 and Supplementary Movie 1).

### Photosynthetic activity of living hydrogels

To assess the photosynthetic activity of the living hydrogels, the $O_2$ microenvironment and $O_2$ turnover were measured in light and darkness using $O_2$ microsensors. $O_2$ evolution was detected from an antibiotic-resistant *S. elongatus* strain WT(Sp$^R$Sm$^R$Gm$^R$), which is used

as a control for engineered strains described in later sections (Supplementary Data 4), embedded in hydrogels (Supplementary Fig. 8a). $O_2$ microsensors revealed an actively photosynthesizing and hyperoxic microenvironment on the surface of the living hydrogels (Supplementary Fig. 8b). For the hydrogel containing WT(Sp$^R$Sm$^R$Gm$^R$) cells, mean $O_2$ concentration adjacent to the hydrogel surface was 346 ± 4.9 μM SE; (Tukey post hoc $p = < 0.01$) at an incident irradiance of 80 μmol photons m$^{-2}$ s$^{-1}$. Net photosynthesis (Fig. 2d) observed for the WT(Sp$^R$Sm$^R$Gm$^R$) strain was 0.031 nmol $O_2$ cm$^{-2}$ s$^{-1}$ (Tukey post hoc, $p < 0.01$). No oxygen evolution was detected from the unloaded hydrogel; however, slight respiratory activity was detected indicative of the presence of low levels of opportunistic bacteria. During darkness hydrogels encapsulating the WT(Sp$^R$Sm$^R$Gm$^R$) strain showed limited respiratory activity and $O_2$ concentrations were close to

**Fig. 3 | YFP fluorescence is inducible in hydrogels containing RiboF-YFP⁺ strains. a** Schematic overview of the genetic circuit used in the construction of RiboF-YFP⁺ strain. **b** Representative fluorescence microscopy images of hydrogels containing cells expressing a YFP reporter. Hydrogels containing the YFP⁺, YFP⁻, and RiboF-YFP⁺ strains were supplemented with either 1% DMSO vehicle control or 1 mM theophylline. Images are shown for the hydrogels under brightfield (BF), TRITC, and YFP channels. Experiments were conducted with independent samples ($n = 3$). Scale bar corresponds to 100 μm. **c** YFP fluorescence measured from the BG-11 medium each ELM structure was incubated. Experiments were conducted with independent biological samples ($n = 3$). *** and ns indicate a $P$ value ≤ 0.001, and not significant, respectively. The $P$ values in **c** are (from left to right) 0.1384, 0.0886, and 0.0002. $P$ values were calculated using a two-tailed Student's $t$ test. Data are mean ± S.D.

ambient seawater values (Supplementary Fig. 8e). Measurements of variable chlorophyll fluorimetry were performed to determine the photosynthetic efficiency of the living hydrogels over a range of irradiance regimes. For the WT(Sp^RSm^RGm^R) *S. elongatus* strain, photosynthesis was saturated between 200 and 250 μmol photons m$^{-2}$ s$^{-1}$ with an irradiance at onset of saturation ($E_k = P_{max}/\alpha$) of ~85 μmol photons m$^{-2}$ s$^{-1}$, indicative of low-light adaptation (Supplementary

Fig. 8d). Electron transport above 250 μmol photons m$^{-2}$ s$^{-1}$ was fully inhibited (Supplementary Fig. 8f).

While hydrogels containing *S. elongatus* could grow from low initial cell densities under low to moderate light, growth was arrested in samples grown under irradiances greater than 300 μmol photon m$^{-2}$ s$^{-1}$. Photosynthesis vs irradiance curves support saturation of photosynthesis at similar irradiance levels for these

low-light adapted hydrogels (Supplementary Fig. 8d). Because solar radiation levels on the earth's surface can reach approximately 2000 µmol photons m$^{-2}$ s$^{-1}$, freshly printed biocomposites will likely suffer in natural high-light environments. However, aquatic environments such as lakes and rivers often contain large concentrations of colored dissolved organic matter that rapidly attenuate light with depth[41], providing a hospitable environment for the ELMs presented here.

## Regulation of YFP expression in hydrogels

To investigate whether cells grown within the hydrogel could respond to an external chemical stimulus, *S. elongatus* was transformed with plasmids pAM4909, pAM5027, and pAM5057 to yield strains YFP$^+$ (constitutive expression YFP positive control), YFP$^-$ (negative control), and RiboF-YFP$^+$ (riboswitch-regulated YFP expression), respectively (Supplementary Data 2, 4). A schematic of the RiboF-YFP$^+$ genetic circuit is illustrated in Fig. 3a; the engineered *S. elongatus* strain responds to the chemical inducer theophylline from the surrounding environment by a conformational change in mRNA bearing Riboswitch-F, which regulates translation of the YFP reporter protein. The YFP$^+$ construct contains a constitutive *con*II promoter driving the expression of YFP. After treatment with 1 mM theophylline in dimethyl sulfoxide (DMSO) or a 1% DMSO vehicle control for 24 hours, YFP expression was determined qualitatively within the hydrogel using fluorescence microscopy (Fig. 3b). A larger version of Fig. 3b is available as Supplementary Fig. 9a–c. No YFP fluorescence was observed from hydrogels containing the YFP$^-$ strain. Constitutive YFP fluorescence was observed from hydrogels containing the YFP$^+$ strain in a medium containing either 1 mM theophylline or 1% DMSO. For hydrogels containing the RiboF-YFP$^+$ strain, growth in 1 mM theophylline produced similar fluorescence levels to hydrogels containing the YFP$^+$ control strain, while no YFP fluorescence was detected in hydrogels supplemented with 1% DMSO. These data demonstrate that the cells embedded within the hydrogel can respond to an external chemical stimulus and be engineered for tight regulation of expression. Experiments conducted with WT strains did not yield any YFP fluorescence from hydrogels when imaged (Supplementary Fig. 10a).

To assess whether viable engineered cells may have leaked from the hydrogel into the surrounding medium, media from 5-day-old hydrogels containing the YFP$^-$, YFP$^+$, RiboF-YFP$^+$, or WT strains were incubated with 1 mM theophylline or 1% DMSO for 24 hours and YFP fluorescence was quantified with a fluorescence plate reader. An ~16.8-fold increase in fluorescence was detected in supernatants from the RiboF-YFP$^+$ samples induced with theophylline compared to the uninduced sample (Fig. 3c). No significant difference was observed between induced and uninduced samples of supernatants from all other hydrogels (Fig. 3c and Supplementary Fig. 10b). The presence of viable cells in the surrounding media may be attributed to a low level of diffusion of cells via mechanical agitation in an aqueous solution throughout the hydrogel and at the hydrogel-medium interface, resulting in a slow release of cells.

## Construction of constitutive laccase-expressing and inducible cell death strains

To create living hydrogels suitable for bioremediation, a strain capable of constitutively producing a laccase enzyme was constructed with an additional feature of inducible cell death. The CotA laccase from *Bacillus subtilis* was chosen, as extensive work characterizing the copper-dependent enzyme has demonstrated its capacity for oxidizing synthetic dyes, and it has been previously expressed in cyanobacteria[35,42–44]. A neutral site 2 (NS2) chromosome-integration plasmid (pAM5825) was constructed that encodes a laccase enzyme (CotA) expressed from the constitutive *con*II promoter (Supplementary Data 2, 4), and the construct was

inserted into the *S. elongatus* chromosome. In response to an observed basal level of cell leakage from the hydrogels over time, the system was engineered for inducible cell death as a mechanism to minimize the potential for contamination and biofouling by the ELM. The overexpression of the native *S. elongatus* gene *Synpcc7942_0766* results in the excision of a prophage from the *S. elongatus* genome, causing cellular lysis. In addition to pAM5825, a neutral site 1 (NS1) chromosome-integration plasmid (pAM5829; Supplementary Data 2) containing a *con*II promoter–theophylline-responsive riboswitch upstream of a copy of the *S. elongatus* gene *Synpcc7942_0766* was constructed and inserted into the *S. elongatus* chromosome at NS1, creating strain Laccase$^+$-riboF-Lysis$^+$, and printed in hydrogels in tandem with a corresponding control strain Laccase$^-$-riboF-Lysis$^-$. Figure 4a illustrates a schematic of the Laccase$^+$-riboF-Lysis$^+$ genetic circuit.

Concentrations of O$_2$ at the hydrogel surface and oxygen evolution and respiration rates of the hydrogels embedded with the Laccase$^+$-riboF-Lysis$^+$ were similar to hydrogels embedded with the WT(Sp$^R$Sm$^R$Gm$^R$) strain (Supplementary Fig. 8a–e; see supplementary information note for details), and genotypic characterization and expression of the 52-kDa CotA enzyme were confirmed (Supplementary Fig. 12a, b).

## ABTS-Laccase activity test for 3D-printed Laccase$^+$-riboF-Lysis$^+$ and control Laccase$^-$-riboF-Lysis$^-$ patterns

Laccase activity was tested against ABTS both within the ELM hydrogel and from the surrounding solution. The hydrogels were first separated from their surrounding BG-11 media and submerged in a reaction buffer mixture with 2 mM ABTS. Following 1 hour of incubation, a set of hydrogels containing Laccase$^-$-riboF-Lysis$^-$ and Laccase$^+$-riboF-Lysis$^+$ saturated with ABTS solution were removed, placed on the surface of sterile agar LB plates, and left to incubate for 4 days (Fig. 4b). Over the 4-day period, a purple coloration developed in the Laccase$^+$-riboF-Lysis$^+$ hydrogel indicative of the oxidation of ABTS, while the hydrogel containing the Laccase$^-$-riboF-Lysis$^-$ strain remained the natural green hue produced by cyanobacterial pigments that was observed in all patterns at time 0.

To quantify the laccase activity of the ELMs in the surrounding solution, a set of hydrogels containing Laccase$^-$-riboF-Lysis$^-$, Laccase$^+$-riboF-Lysis$^+$, and unloaded gels were submerged in a buffer solution containing ABTS and incubated for 4 days with daily sampling for quantification of the oxidation of ABTS in the reaction buffer (Fig. 4c). After 24 hours, 71.5 ± 13.6 nmol/ml of ABTS was oxidized by the Laccase$^+$-riboF-Lysis$^+$ hydrogel. A small but significant level of activity against ABTS was observed in the Laccase$^-$-riboF-Lysis$^-$ relative to the control unloaded gel. Over the course of 4 days of incubation, the level of oxidized ABTS from the hydrogel embedded with Laccase$^+$-riboF-Lysis$^+$ increased to 172.0 ± 21.3 nmol/mL, which was significantly greater than either the Laccase$^-$-riboF-Lysis$^-$-containing or unloaded gels (*P* value < 0.01). The oxidation of the ABTS substrate to its cationic radical form was also evident from the dark green coloration observed in the hydrogels bearing strain Laccase$^+$-riboF-Lysis$^+$ over time (Supplementary Fig. 13).

To assess whether the CotA enzyme escapes from the hydrogel, either by means of secretion or cell lysis within the gel, the laccase activity of the supernatant of media surrounding hydrogels was measured from samples taken five days post-printing (Supplementary Fig. 14). The activity observed from the supernatant surrounding the Laccase$^+$-riboF-Lysis$^+$ containing hydrogels was significantly greater than either the Laccase$^-$-riboF-Lysis$^-$ containing or unloaded gels. However, a low level of ABTS oxidation was observed in supernatants from hydrogels embedded with Laccase$^-$-riboF-Lysis$^-$ relative to unloaded hydrogels at each timepoint, indicating that the secretion or lysis products of the control Laccase$^-$-riboF-Lysis$^-$ strain also confer some level of oxidative activity against the substrate ABTS, albeit a

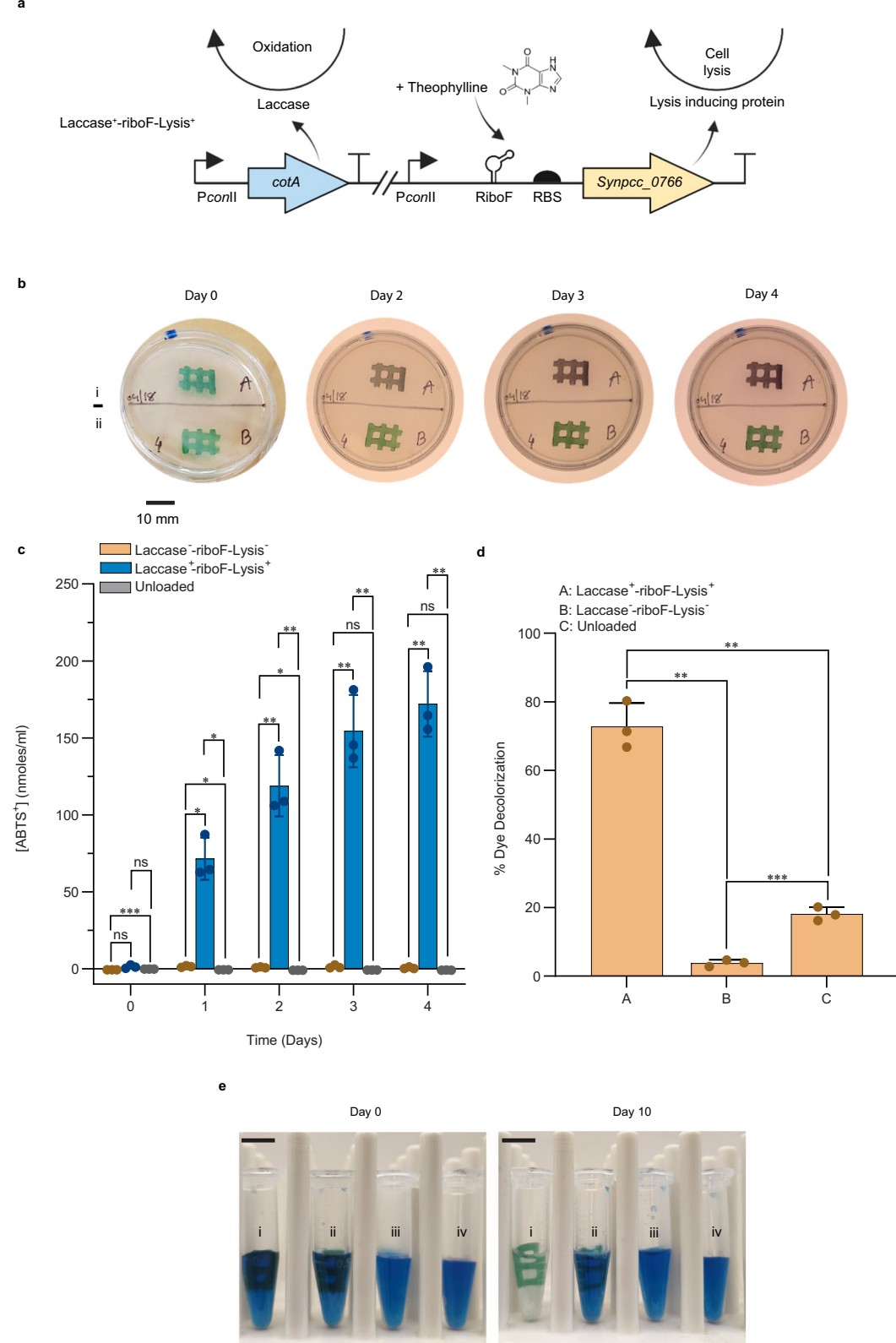

reduced level relative to the hydrogels embedded with Laccase+-riboF-Lysis+. Additionally, a strain in which the expression of laccase is regulated by Riboswitch-F (RiboF-Laccase+) was constructed, printed within hydrogels, and tested for laccase activity with and without the addition of theophylline (see Supplementary Fig. 11 and Supplementary Note).

**Indigo carmine oxidation activity for 3D-printed Laccase+-riboF-Lysis+ and control Laccase−-riboF-Lysis− patterns**

To assess whether hydrogels containing the engineered strains of Laccase+-riboF-Lysis+ can serve as a tool for bioremediation, the hydrogels were assayed for their ability to decolorize a common textile dye pollutant, indigo carmine, a capability previously demonstrated

**Fig. 4 | Hydrogels containing Laccase⁺-riboF-Lysis⁺ strain have enhanced oxidative activity. a** Schematic overview of the genetic circuit used in the construction of Laccase⁺-riboF-Lysis⁺ strain. **b** Visual assay of hydrogel ABTS oxidation activity over four days. (i) Subsection of a hydrogel containing strain Laccase⁺-riboF-Lysis⁺. (ii) Subsection of a hydrogel containing the Laccase⁻-riboF-Lysis⁻ control strain. The scale bar on the bottom left corresponds to 10 mm. Experiment was repeated thrice with similar results based on independent samples. **c** Time-course of the oxidation of ABTS in reaction buffer after the addition of hydrogels containing either the Laccase⁻-riboF-Lysis⁻ control strain, Laccase⁺-riboF-Lysis⁺ strain, or an unloaded hydrogel control. Experiments were conducted in triplicate from independent samples, $n = 3$. *, **, ***, and ns indicate $P$ values ≤ 0.05, 0.01, 0.001, and not significant, respectively. $P$ values were calculated using a two-tailed Student's $t$ test. The $P$ values in **c** (from left to right) are 0.0975, 0.0001, 0.0864, 0.0123, 0.0232, 0.0117, 0.0093, 0.0153, 0.0091, 0.0077, 0.0742, 0.0075, 0.0051, 0.0700, and 0.0050. Data are mean ± S.D. **d** Percent decolorization of indigo carmine (0.1 mg/mL in BG-11) by hydrogels printed with either Laccase⁻-riboF-Lysis⁻ or Laccase⁺-riboF-Lysis⁺ strains or unloaded hydrogel after incubation for 10 days. Experiments were conducted in triplicate from independent samples, $n = 3$. *, **, ***, and ns indicate $P$ values ≤ 0.05, 0.01, 0.001, and not significant, respectively. $P$ values were calculated using a two-tailed Student's $t$ test. The $P$ values in **d** (from left to right) are 0.0033, 0.0056, and 0.0004. Data are mean ± S.D. **e** Representative images of hydrogels at incubation time 0 (left) and after 10 days of incubation (right). (Roman numerals indicate the addition of i) hydrogel printed with Laccase⁺-riboF-Lysis⁺, ii) hydrogel printed with Laccase⁻-riboF-Lysis⁻, iii) unloaded hydrogel, iv) BG-11 with dye only.) Scale bars in upper left correspond to 10 mm. Experiment was repeated thrice based on independent samples.

for a heterologously-expressed laccase[35]. Five-day-old hydrogels containing either Laccase⁺-riboF-Lysis⁺, Laccase⁻-riboF-Lysis⁻, or an unloaded-control were submerged in a BG-11 solution containing 0.1 mg/ml of indigo carmine. The decolorization of indigo carmine was subsequently monitored over the course of 10 days and quantified by proxy of absorption at 612 nm, the absorption maximum of indigo carmine in the visible spectrum (Fig. 4d, e). The hydrogel samples embedded with Laccase⁺-riboF-Lysis⁺ had an average decolorization of 72.8 ± 6.8%, while the Laccase⁻-riboF-Lysis⁻ and unloaded hydrogel had decolorization averages of 3.8 ± 1.0% and 18.1 ± 2.0%, respectively. The percent dye decolorization in hydrogels containing Laccase⁻-riboF-Lysis⁻ and unloaded hydrogels is due to the adsorption of indigo carmine by the alginate polymer matrix, as has been previously reported for other substrates[45,46], and is observable as the blue hue of unloaded hydrogels removed from the solution (Supplementary Fig. 15).

The hydrogel embedded with Laccase⁺-riboF-Lysis⁺ successfully decolorized ~70% of the indigo carmine over the course of 10 days with the added benefit of being physically removable from the system without requiring energy-intensive steps to clarify a liquid culture, such as centrifugation. The unloaded hydrogel also produced a significant level of apparent indigo carmine decolorization due to the absorption of indigo carmine by the hydrogel matrix itself. Interestingly, the unloaded hydrogel consistently had a greater level of decolorization due to non-specific adsorption of the dye molecule than did hydrogels containing the Laccase⁻-riboF-Lysis⁻ control strain.

## Inducible cell death of cyanobacteria

When designing ELMs, consideration of the unintended consequences of contamination from the biological component into the surrounding environment should be taken into consideration. The *Synpcc7942_0766* gene, when overexpressed, results in the excision of a defective prophage from the *S. elongatus* genome[47], resulting in cellular lysis (Fig. 5a). Hydrogels containing Laccase⁺-riboF-Lysis⁺ or Laccase⁻-riboF-Lysis⁻ (control) strains were transferred to fresh medium after 5 days of initial growth and incubated for an additional 12 days in fresh growth media supplemented with either 1 mM theophylline in DMSO, 1% DMSO, or medium only (Fig. 5b). After 12 days of growth, visual inspection indicated that accumulation of biomass in the surrounding medium was absent in the flask containing the Laccase⁺-riboF-Lysis⁺ strain supplemented with 1 mM theophylline. The timescale of response to induction of cell death in the surrounding medium was assessed by supplementing fresh hydrogel supernatants containing cells of either Laccase⁺-riboF-Lysis⁺ or Laccase⁻-riboF-Lysis⁻ strains with either 1 mM theophylline or 1% DMSO and monitoring growth of the cultures for four days (Fig. 5c). Within 48 hours, ectopic induction of Synpcc7942_0766 in the Laccase⁺-riboF-Lysis⁺ strain supplemented with 1 mM theophylline resulted in a decrease in $OD_{750}$, initially visible as clumps of senescent cell matter that continued to break down to produce a pale hazy medium, indicating death of the culture (Fig. 5c, d). Cultures of Laccase⁺-riboF-Lysis⁺ supplemented with 1% DMSO and

Laccase⁻-riboF-Lysis⁻ supplemented with either 1 mM theophylline or 1% DMSO continued to grow at similar rates to one another, demonstrating the utility of engineered inducible cell death in an ELM as a mechanism of targeted reduction of biological contamination by the material.

Riboswitches are increasingly being used for applied bioengineering due to the simplicity of ligand-aptamer interactions[13,48,49]. The use of riboswitches to regulate protein expression within hydrogels was successful for YFP, laccase, and the lysis-inducing protein and should function well in future biocomposites that contain alternative cyanobacterial or algal strains. We attribute the success of regulation by riboswitches to the diffusion of the small molecule theophylline throughout the porous alginate hydrogel.

## Discussion

We optimized the composition of an alginate-based hydrogel for 3D printing and viability of the cyanobacterium *S. elongatus*, and engineered several strains of the cyanobacterium to be stimulus-responsive and produce functional outputs while embedded within the hydrogel. Previously, hydrogels constructed with cyanobacteria as a biological component[36,50,51] have been limited to wild-type strains. In this study, we leveraged genetic toolkits for synthetic biology to produce photosynthetic ELMs with functional outputs and tailored regulatory circuits. The use of riboswitches to regulate gene expression within the living material yielded stimulus-responsive materials with the capacity for dye decolorization and inducible cell death to prevent cellular contamination in the environment.

While the photosynthetic ELMs designed in this study have functional outputs suitable for bioremediation, numerous improvements could make the material more appropriate for use outside of the laboratory setting. Notably, improvements in enzymatic activity per unit volume of hydrogel will be necessary to move from the laboratory scale to large-scale applications, such as bioremediation in lakes or water treatment plants. One strategy for the improvement of activity from the material could be the modification of protein products to be tagged for secretion from the cell resulting in higher concentrations in solution and allowing for the production of bioproducts that can be toxic to the cells at high intracellular concentrations. Knowledge of mechanisms and systems for the secretion of larger recombinant proteins in cyanobacteria such as *S. elongatus* PCC 7942 will greatly benefit the output potential of cyanobacterium-based ELMs. Given the complexity of the composite material system, we were unsuccessful in identifying the localization of the CotA protein in the ELM. There are several potential scenarios under which catalysis could occur, including intracellular catalysis in which the substrate and/or mediator molecules cross into the cell and the product is exported or diffuses out of the cell, extracellular catalysis in which the laccase is exported from the cell, extracellular catalysis by a stable laccase following cell lysis, or likely a combination of the above. Structurally, factors such as the ratio of the β-D-mannuronic (M) and α-L-guluronic (G) acid (M/G) in the alginate, type of crosslinking (e.g., chemical or ionic), and/or the

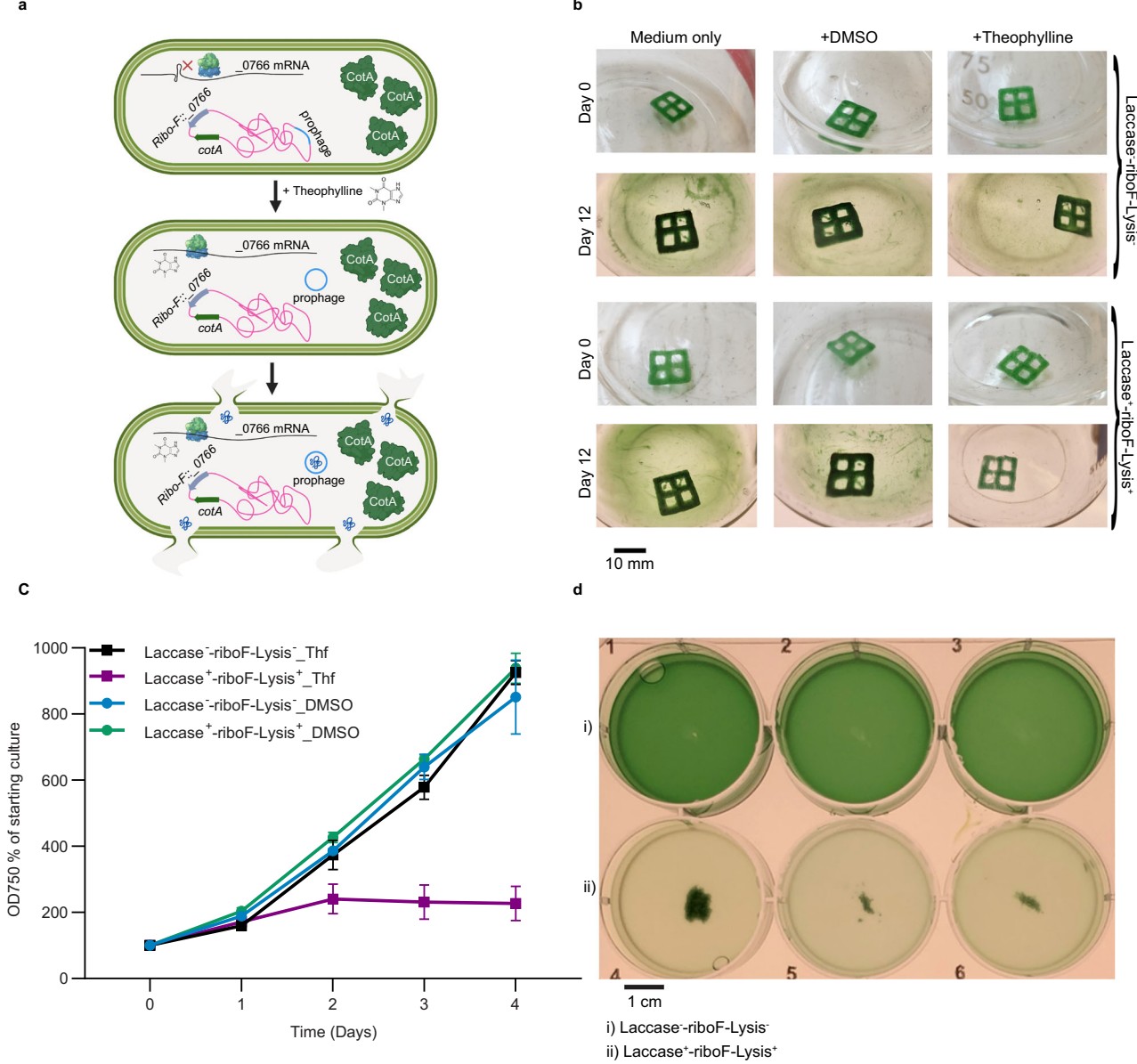

**Fig. 5 | Inducible cell death of strain Laccase⁺-riboF-Lysis⁺. a** Schematic of constitutive expression of CotA, and riboswitch conformation change with the addition of theophylline, leading to the overexpression of Synpcc7942_0766, prophage excision, and subsequent cell death. **b** Representative images of day 0 and day 12 images of the hydrogels containing Laccase⁻-riboF-Lysis⁻ control and Laccase⁺-riboF-Lysis⁺ strains supplemented with either medium only, 1% DMSO, or 1 mM theophylline. The experiment was repeated thrice with similar results. **c** Growth curves of the Laccase⁻-riboF-Lysis⁻ control and Laccase⁺-riboF-Lysis⁺ strains with the addition of either 1 mM theophylline (Thf) (square) or 1% DMSO (circle) in solution. Experiments were conducted in triplicate from independent samples ($n = 3$). Data are mean ± S.D. **d** Representative image of liquid cultures of Laccase⁻-riboF-Lysis⁻ and Laccase⁺-riboF-Lysis⁺ strains three days post supplementation with 1 mM theophylline. Experiments were conducted in triplicate from independent samples ($n = 3$).

concentration of divalent ions significantly affects the structural integrity and the degree of swelling of the hydrogel network. The presence of the divalent ions in the surrounding environment plays a crucial role in the crosslinking process and may result in the weakening of the gel if $Ca^{2+}$ ions are substituted[52,53].

Careful consideration should be paid to the influence of materials containing biological components on the natural environment, such as biofouling, and subsequent effects on ecological systems. The ELMs presented here address this issue by incorporating a mechanism for inducible cell death by the addition of the small molecule theophylline. While theophylline is a natural product present in small quantities in tea and cocoa, it is not native to aquatic systems and its use in the natural environment could pose a risk of unintended chemo-ecological consequences. Alternative

regulatory circuits with more compatible inductive signals could be engineered for use in the natural environment. For example, spectrally responsive photoreceptors or temperature-sensitive signaling transduction pathways could allow for diurnal or seasonal cell death of biological contamination.

The ELM presented here serves as a proof-of-concept for the engineering and incorporation of cyanobacteria into a polymeric matrix to produce stimuli-responsive materials with functional outputs of real-world utility. Utilizing cyanobacteria as the biological component, these ELMs fundamentally depend on only light, $CO_2$ as a carbon source, and minimal nutrients for survival. While the biocomposites demonstrated here have been designed for use in bioremediation, the functional outputs of cyanobacterial ELMs could be engineered to be broadly impactful.

## Methods

### Cultivation of cyanobacteria

Wild-type control and genetically engineered *S. elongatus* PCC 7942 cyanobacterial cells were cultivated in BG-11 medium[54] in conical flasks and grown at 30 °C while shaking at 120 rpm under continuous illumination of 75 μmol photons $m^{-2} s^{-1}$. Irradiance measurements were conducted with a 4π quantum light meter (model QSL-100, Biospherical Instruments). BG-11 was supplemented with 2 μg/ml spectinomycin, 2 μg/ml streptomycin, and/or 2 μg/ml gentamycin for selection as required. Unless otherwise noted, cells were grown for 5–6 days from an initial $OD_{750}$ of 0.05 to the mid-log phase ($OD_{750} = 0.5$–0.6) prior to the preparation of bioinks.

### Preparation of alginate ink and bioink

Alginate hydrogels were prepared using a mixture of sodium alginate with $CaSO_4$ and $CaCl_2$ as crosslinking agents at optimized concentrations. In brief, a 4% w/v sodium alginate solution was prepared by dissolving sodium alginate powder (Sigma-Aldrich) in autoclaved deionized water. The solution was continuously stirred overnight at room temperature, achieving a clear, viscous solution. Stock solutions of $CaSO_4$ and $CaCl_2$ were prepared in autoclaved deionized water at concentrations of 50 mM and 100 mM, respectively. All solutions were stored at 4 °C until use. A hydrogel ink solution was prepared for printing by mixing the alginate solution and $CaSO_4$ in a 2:1 ratio. The slurry suspension of 50 mM $CaSO_4$ was mixed thoroughly (25–30 times) with alginate solution using an in-house fabricated 3-way syringe mixer and then transferred to a 3 cc syringe for 3D printing. The ink in the 3 cc syringe was placed within a 50 mL falcon tube and centrifuged at $2200 \times g$ for 10 min to remove any air bubbles prior to printing.

Bioinks were prepared with cells from cultures at an $OD_{750}$ of 0.5–0.6. The cells were collected by centrifugation at $3500 \times g$ for 5 min. The pellet was mixed in the plotting paste using the 3-way syringe mixer (Supplementary Fig. 16). All glassware used in the process was pre-sterilized by autoclaving and all accessories required for cell mixing and 3D printing were pre-sterilized by UV sterilization. Both the ink and the bioink were prepared fresh prior to each experiment.

### 3D printing of alginate ink and bioink

$CaSO_4$ was used to partially crosslink the alginate chains to make the gel suitable for direct-ink-writing (DIW) printing. Various combinations of alginate solutions (0.5–5% w/v) and $CaSO_4$ (25-100 mM) concentrations were tested at different pressures (50–350 kPa) using steel blunt end syringe needles (22, 27, and 32-gauge sizes) to optimize conditions for printability. The homogeneous mixture that gave the most stable structure at a given needle size and pressure was used for printing the bioink. In brief, the ink with or without cyanobacterial cells was loaded in a 3 cc printing syringe barrel, equipped with a 27-gauge needle. The ink was 3-D printed with a pre-designed geometry using the Inkredible+ Bioprinter (Cellink, USA). The printing parameters were as follows: pressure -200 kPa, infill density 50%, fill pattern rectilinear, and speed 80 mm/sec. All prints were produced in ambient conditions at room temperature. The printed geometries were designed using CAD software and converted to Gcode using the Cellink Heartware v2.4.1. software. Ink without *S. elongatus* cells was used as a cell-free control hydrogel in all experiments. Post printing, the multilayered 3D-printed structures were immersed in 100 mM $CaCl_2$ for 15 min to allow the patterns to be fully cross-linked and then incubated in 47 mm petri dishes containing 5 ml BG-11 medium supplemented with antibiotics. Unless otherwise mentioned, all the scaffolds with either WT or engineered strains were kept in an incubator at an irradiance level of 20 μmol photons $m^{-2} s^{-1}$ and 28 °C for initial growth prior to experimentation.

### Rheological characterization and viscosity measurements

Viscoelastic moduli and viscosity measurements of the alginate ink with and without the inclusion of WT cells were carried out using a Discovery HR-30 Hybrid Rheometer (TA Instruments). Data was acquired using the TRIOS Software. Both measurements were conducted using a 20 mm parallel plate geometry in a Peltier plate setup. For amplitude sweep measurements, angular frequency was 10 rad/sec and strain ranged from 0.01% to 1000.0%. For viscosity, flow sweep was measured at a continuous shear rate of $0.01 s^{-1}$ to $1000.0 s^{-1}$. All characterizations were conducted at a temperature of 25 °C.

### Growth optimization of the 3D-printed WT *S. elongatus*

The growth of WT *S. elongatus* cells was monitored and optimized in different geometries of 3D-printed patterns over time. Specifically, cells encapsulated in three different printed structures (disk, honeycomb, and grid_A) were incubated in BG-11 medium with constant illumination for 7 days and were visually monitored for color density over time. Images were taken at regular time intervals.

### Cell viability of 3D-printed WT *S. elongatus*

Cell viability of WT *S. elongatus* within the 3D-printed patterns was assessed using confocal microscopy. Live/dead cell assays were performed by imaging chlorophyll autofluorescence and SYTOX Blue (SB; Thermo Fisher Scientific) stain fluorescence. 3D-printed grid_A patterns were cut to yield two interconnected hollow squares under sterile conditions at regular time intervals and incubated in 5 μL of a 1 mM SB stock solution in 1 mL of BG-11 medium for 5 min in the dark. The effective concentration of the stain was 5 μM. After incubation, the samples were washed three times with BG-11, and images were recorded and analyzed on a Leica Sp8 confocal microscope with lightning deconvolution and white light laser, using the LAS X software. The channel for SB detection was set at excitation 405 nm, and the emission was collected at 450–510 nm. The channel for chlorophyll autofluorescence was set at excitation 488 nm and the emission was collected at 685–720 nm. Samples from different time points were imaged to monitor cell viability. Independent regions of the hydrogel were used for imaging of each timepoint to minimize contamination. For cell viability analysis three confocal images from each timepoint from independent samples ($n = 3$) were taken. Raw data were processed in image processing software Fiji[55] using the JACoP[56] colocalization plugin. The fraction of overlapping channel intensities was calculated using Mander's Coefficients (using a threshold value of 28 for each channel). The live/dead cell percentage was calculated based on the obtained value. Z-stack image acquisition was performed for the construction of 3D images. A total of 69 confocal images were taken from the hydrogel containing WT *S. elongatus* with $x$ and $y$ axes physical lengths of 289.21 μm and a $z$ axis physical length of 51.69 μm. 3D images were reconstructed in Imaris Viewer v9.9.0.

### Field emission scanning electron microscopy

Electron microscopy images of multiple regions of freeze-dried hydrogel samples, printed with and without WT cells, were taken on an FEI Apreo FESEM (Thermo Scientific). Sample subsections were cut and stored at −80 °C for 24 hours. The samples were subsequently lyophilized to remove residual water, and the dried samples were prepared for imaging. The samples were adhered to SEM sample stubs using double-sided carbon tape and were coated uniformly with gold particles for 60 msec. Images were taken with an ETD detector at 10-20 kV, using the xT microscope control v13.9.1.

### $O_2$ microsensor measurements

Clark-type $O_2$ microsensors (tip size 25–50 μm, Unisense, Aarhus, Denmark) were used to measure the $O_2$ microenvironment and $O_2$ turnover of the living hydrogels in light and in darkness. Microsensor

measurements were performed as described previously[57,58]. Briefly, $O_2$ sensors were mounted on a motorized micromanipulator (MU1, PyroScience GmbH, Germany) that was attached to an optical table. Cell-loaded and unloaded hydrogels were placed in a custom-made acrylic system that provided slow laminar flow at a rate of approximately 1 cm/s. $O_2$ microsensor measurements were performed at the cross-section between the printed horizontal and vertical areas. Preliminary $O_2$ mapping suggested that these are the areas of highest photosynthetic activity. Microsensor profiles were generated by carefully positioning the sensor at the surface of the hydrogel with the aid of a digital microscope (Dino-Lite, US) and $O_2$ values were logged using Unisense software (Sensor Trace Suite (Logger v3.3)). $O_2$ profiles were measured from the hydrogel surface through the diffusive boundary layer into the overlying water column in steps of 50–100 $\mu$m using the motorized micromanipulator that was operated by the manufacturer's software (Pyroscience software (Profix v4.6)). Net photosynthesis was determined at an incident downwelling irradiance of 80 $\mu$mol photons $m^{-2} s^{-1}$ as provided by a fiber optic halogen light source (ACE, Schott GmbH). Dark respiration was calculated as the diffusive $O_2$ flux in darkness. The diffusive $O_2$ flux was calculated using Fick's first law of diffusion as described previously[57] using a diffusion coefficient of $DO_2 = 2.2186 \times 10^{-5}$. The index of light adaptation (i.e., the irradiance at onset of saturation) $E_k$ was calculated as $E_k = P_{max}/\alpha$; where $P_{max}$ = maximal photosynthesis rate and $\alpha$ = the initial slope of the light curve indicating light use efficiency. Data was plotted using OriginPro v2021.

### YFP monitoring by fluorescence microscopy from cell-laden hydrogel

YFP fluorescence from the cells within the hydrogel matrix was qualitatively measured by capturing images of hydrogel patterns on a Nikon Eclipse Ni fluorescence microscope equipped with a Nikon DS Qi2 camera at 20X or 40X magnification. 3D-printed cell-laden grids were incubated in media for 5 days, removed, and fresh 3 ml media supplemented with either 1 mM theophylline or a 1% DMSO vehicle control was added to the gel. After 24 hours, the patterns were removed from the solutions and a subsection of each gel was cut for imaging. Brightfield, chlorophyll autofluorescence, and YFP fluorescence images were captured for each sample containing the different YFP strains. A TRITC channel was used to monitor in vivo chlorophyll autofluorescence from cyanobacteria. Images were analyzed using the NIS-Elements Viewer v5.21.00.

### YFP monitoring by fluorescence measurements from surrounding medium

To assess the presence of YFP in the surrounding medium, the 24-hour theophylline/DMSO-supplemented samples (as prepared above) were analyzed using fluorescence measurements. The surrounding media of the hydrogel was collected and centrifuged at 4500 × g for 5 min and the emission intensity of YFP from the supernatant was measured in a 96-well plate with a Tecan Infinite M200 plate reader (TECAN). The excitation and emission wavelengths were set for YFP to excitation 490/9 nm and emission 535/20 nm. Measurements were taken from hydrogels containing 3 independent clones of each YFP strain along with the WT cell-loaded sample.

### SDS-PAGE of CotA

Cultures of WT and Laccase+-riboF-Lysis+ were grown in liquid culture under 75 $\mu$mol photons $m^{-2} s^{-1}$ with continuous shaking at 30 °C to an $OD_{750}$ of 0.5. 10 mL of cultures were pelleted by centrifugation at 4500 × g and aspirated. The pellets were then resuspended in 1 mL of cold lysis buffer (50 mM Tris-HCl pH 8.0, 150 mM NaCl) supplemented with 1 mM PMSF and Complete Mini Protease Inhibitor (Roche). The samples were then homogenized in a BeadBlaster 24 R (Benchmark Scientific) at 4 °C. The crude protein was then

clarified by centrifugation at 4 °C for 20 min at 18,000 × g, after which the crude protein concentration was determined using a Pierce Coomassie Plus Protein Assay (Bradford), with bovine serum albumin used as a standard. After boiling the sample with 2× Laemmli Sample Buffer, 30 $\mu$g of sample was loaded into a Mini-PROTEAN® TGX Precast Gel (Bio Rad) and run at 150 V. The gel was then incubated in InstantBlue Coomassie Protein Stain (Abcam) overnight prior to imaging.

### ABTS activity assays

Laccase activity of either the hydrogel or the surrounding solution was measured using ABTS (Thermo Fisher Scientific). For assays involving the induction of RiboF-Laccase+, an ~83 mm³ subsection of a hydrogel (two interconnected hollow squares) was added to 1 mL of BG-11 in a 2 mL Eppendorf tube and supplemented with either 1 mM theophylline (Sigma-Aldrich; 100 mM stock dissolved in 100% DMSO) or 1% DMSO as control and allowed to incubate for 3 days under 50 $\mu$mol photons $m^{-2} s^{-1}$ prior to conducting the ABTS assays. The hydrogel sample was then transferred to a 1 mL reaction buffer mixture of 100 mM sodium acetate buffer, 3.16 $\mu$M $CuSO_4$, pH 5.0, and 2 mM ABTS in a 1.5 ml conical microtube and incubated in darkness at 28 °C. The oxidation of ABTS was determined by measuring the absorbance of the reaction mixture at 420 nm[59] after 24 hours. A blank reading at 420 nm of the reaction mixture without a hydrogel was subtracted from all samples.

For ABTS assays within the hydrogels with Laccase+-riboF-Lysis+ cells, an ~83 mm³ subsection of the hydrogel was submerged in 1 ml of reaction buffer mixture (as mentioned above) and incubated for 1 hour. The hydrogel ELMs were then removed from the solution and placed on sterile agar LB plates and incubated for 4 days in a dark environment at 28 °C. Photographs were taken at regular time intervals. For laccase activity in the surrounding solution, similar subsections of the hydrogels were submerged in 1 ml of the reaction buffer (as mentioned above) containing 2 mM ABTS and incubated for 4 days in darkness at 28 °C. The absorbance of the reaction mixture was measured at 420 nm at regular time intervals, relative to the absorbance at 0 h. A blank reading at 420 nm consisting of the reaction mixture without a hydrogel was subtracted from all samples. Additionally, for the Laccase+-riboF-Lysis+ hydrogel set, supernatant laccase activity was measured. 1 mL of the medium surrounding a 5-day grown hydrogel ELM was centrifuged for 5 minutes at 4500 × g. 500 $\mu$L of the supernatant was combined with 460 $\mu$L 200 mM sodium acetate buffer, pH 5.0, and 40 $\mu$L of a 50 mM stock of ABTS for a final concentration of 100 mM sodium acetate buffer, pH 5.0, and 2 mM ABTS. The samples were incubated in darkness at 28 °C, and oxidation of ABTS was measured at regular time intervals, relative to the absorbance at 0 h. A blank reading at 420 nm consisting of the BG-11 in the reaction mixture was subtracted from all supernatant samples. For all the assay measurements, 200 $\mu$L of samples were taken in a 96-well plate and measured with a Tecan Infinite M200 plate reader (TECAN).

### Indigo carmine decolorization assays

The decolorization of indigo carmine by hydrogels was determined by suspending an 83 mm³ subsection of a hydrogel (two interconnected hollow squares) in a 500 $\mu$L solution of BG-11 media with 0.1 mM indigo carmine (Sigma-Aldrich). Indigo carmine concentrations were determined spectrophotometrically by generating a standard curve for absorbance at 612 nm for 200 $\mu$L samples in a 96-well plate with a Tecan Infinite M200 plate reader. The indigo carmine hydrogel mixture was incubated in darkness, and the absorbance of 200 $\mu$L samples was measured at 612 nm at time 0, and intermittently till 10 days of incubation at 28 °C. The percentage of dye decolorization after 10 days was determined relative to absorbance at 612 nm at time 0 for a given sample.

## Inducible cell death by phage-gene lysis within 3D-printed hydrogel

Small grid_B hydrogel patterns with strains Laccase⁻-riboF-Lysis⁻ and Laccase⁺-riboF-Lysis⁺ were printed and incubated for 5 days of growth. After 5 days, the constructs were transferred to flasks containing 5 ml of either BG-11 medium, 1 mM theophylline (prepared from a 100 mM stock dissolved in 100% DMSO), or 1% DMSO as a control. The flasks were kept under 75 µmol photons $m^{-2} s^{-1}$ with continuous shaking at 30 °C. Images of the experimental samples were captured on day 0 and day 12.

## Inducible cell death by phage-gene lysis in liquid culture

Strains Laccase⁻-riboF-Lysis⁻ and Laccase⁺-riboF-Lysis⁺ were grown in liquid culture under 75 µmol photons $m^{-2} s^{-1}$ with continuous shaking at 30 °C to an initial $OD_{750}$ of 0.2–0.3, prior to being transferred to six-well plates in 8 ml aliquots. Samples were then induced with either 1 mM theophylline (prepared from a 100 mM stock dissolved in 100% DMSO) or 1% DMSO as a control. The $OD_{750}$ of the cultures was measured daily for four days following induction.

## Construction of plasmids and engineered strains

To create plasmid pAM5823, pAM4950 was digested with SwaI (NEB) to release the *ccdB* toxic gene, and the linearized plasmid was re-ligated with a Quick Ligase Kit (NEB) following manufacturer instructions. Plasmid pAM5825 was created by digesting pAM4950 with SwaI and performing a Gibson Assembly (NEB) between the linearized product and a *cotA* gBlocks Gene Fragment synthesized by Integrated DNA Technologies (IDT) containing a PconII promoter and RBS upstream of the codon-optimized *cotA* sequence. For Gibson Assembly, the *cotA* gBlocks gene fragment contained 20 base pair ends of homology with the linearized pAM4950 product and a *con*II promoter and ribosomal binding site sequence derived from pAM4909 upstream of the *cotA* gene. The *cotA* sequence was derived from a reverse-translation of the protein sequence of CotA from *Bacillus subtilis* (accession number "WP_003243170.1") and was codon optimized for *S. elongatus*. pAM5826 was created by digesting pAM5051 with EcoRI (NEB), and performing a Gibson Assembly between the linearized product and the PCR product of pAM5825 with primer pair cotA_5051-F/cotA_5051-R. pAM5829 was created by digesting pAM4940 with SwaI and performing a Gibson Assembly of the linearized product and the PCR product of *S. elongatus* gDNA and the primer pair 0766_riboF_pAM4940-F/0766_riboF_pAM4940-R targeting the amplification of *Synpcc7942_0766*. The sequence of all plasmids constructed in this study was verified by Sanger sequencing. Complete lists of all plasmids, their respective sequence, and primers used in this study are included in Supplementary Data 2, and 3 respectively.

Plasmids pAM4909, pAM5027, and pAM5057 were transferred into *S. elongatus* by biparental conjugation on selective media[60]. Transformation of *S. elongatus* with all other plasmids was achieved using standard protocols[61]. All *S. elongatus* transformations were performed using WT strain AMC06. *S. elongatus* transformations resulting in strains harboring replicating plasmids were performed using strain AMC2663 from the Golden lab collection, in which RSF1010-based plasmids can replicate. Genotyping of *S. elongatus* strains expected to contain integration into a chromosome-neutral site (not including YFP⁺, YFP⁻, and Ribo-YFP⁺ which bear self-replicating extra-chromosomal plasmids) was performed using colony PCR with Q5 DNA Polymerase (NEB), using primer pair NS1_Screen-F/NS1_Screen-R for transformations in neutral site 1 (NS1), and NS2_Screen-F/NS2_Screen-R for transformation in neutral site 2 (NS2). A complete list of strains used in this study is included in Supplementary Data 4.

## Statistics and reproducibility

Data are presented as the mean ± standard deviation unless noted otherwise. *P* values were calculated using a two-tailed Student's *t* test. The definition of significance was set a priori to $p < 0.05$. Within figures, asterisks represent: $*P < 0.05$; $**P < 0.01$; $***P < 0.001$; $****P < 0.0001$.

No statistical method was used to predetermine sample size. Sample size of $n = 3$ independently transformed clones for all assays was chosen according to convention and due to the high reproducibility of results. No data were excluded from analysis.

## Reporting summary

Further information on research design is available in the Nature Portfolio Reporting Summary linked to this article.

## Data availability

All data supporting this study are included in this article, Supplementary Information, Supplementary Data, and Source Data files. Source data are provided with this paper.

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

## Acknowledgements

We thank Jack Reddan for assistance with the construction of pAM5825, Marie Adomako for the construction of pAM5610, Bryan Bishé and Arnaud Taton for assistance with the conjugation of strains YFP+, YFP–, and RiboF-YFP+, Ryan Simkovsky for consultation

regarding the design of strain Laccase⁺-riboF-Lysis⁺, Nathan Soulier for assistance with the genotyping of strains RiboF-YFP⁺ and Laccase⁺-riboF-Lysis⁺, and David Wirth for assistance with the creation of the 3-way syringe mixer. This work was primarily sponsored by the UC San Diego Materials Research Science and Engineering Center (UCSD MRSEC), supported by the National Science Foundation Grant DMR-2011924. The authors acknowledge the use of facilities and instrumentation supported by the National Science Foundation through the UCSD MRSEC DMR-2011924, the Department of NanoEngineering Materials Research Center (NE-MRC), the UCSD School of Medicine Microscopy core facility funded by NINDS Grant P30 NS047101, and the Gordon and Betty Moore Foundation Aquatic Symbiosis Model Systems (Grant 9325 to D.W. and S.C.). The use of microscopy core equipment was supported by Jennifer Santini and Marcella Erb. Figs. 1 and 5a were created with BioRender.com.

## Author contributions

D.D. and E.L.W. equally contributed to the writing of the manuscript and are co-first authors. D.D. formulated and optimized the bioink, designed and performed 3D printing of samples, conducted rheology and viscosity measurements, conducted growth optimization, cell viability measurements, electron microscopy, and the analysis of data. E.L.W. designed, optimized, and engineered *S. elongatus* strains, performed SDS PAGE, and analyzed data. D.D. and E.L.W. optimized and performed YFP fluorescence assay, ABTS activity assay, indigo carmine decolorization assay, and the inducible cell death assay. D.W. performed $O_2$ microsensor measurements, analyzed data, and assisted in writing the manuscript. E.H. contributed to the design and 3D printing of the samples. D.W., E.H., and S.C. provided feedback and comments on the manuscript. S.S.G., J.G., and J.K.P. supervised the project and edited the manuscript.

## Competing interests

The authors declare no competing interests.
