## [Peer Review file · Nature Communications]

REVIEWER COMMENTS

Reviewer #1 (Remarks to the Author):

Summary:

The manuscript describes the development of Engineered Living Material comprising an alginate scaffold containing photosynthetically active cyanobacteria capable of constitutive/induced laccase enzyme production and induced cell lysis. The proposed light and stimuli-responsive biomaterial is an exciting solution for the bio-contained bioremediation of textile dyes and could be explored further for even more interesting applications with more sophisticated synthetic biology circuits. Overall it is an exciting manuscript at the technical interface of synthetic biology and material science, focusing on the topics of biocatalysis and bioremediation.

An improved manuscript will be attractive to people working in numerous, diverse fields.

The conclusions of the manuscript are supported by the data.

The reference list could be improved. For example, it lacks foundation work on CotA laccase by L.O. Maritns group, and on laccases more broadly.

The manuscript is well written and, overall, clearly presents the data. However, the methodology section of the work requires significant improvement to meet the standard required by Nature Communications. Please see the following recommendations below:

Major issues:

The construction of plasmids

While the overall plan of plasmid construction is understandable and could be approximated by someone skilled in the art, the experimental description presented in the current version of the manuscript is insufficient to replicate the experiment fully. Therefore, the section should be corrected in the revised manuscript to allow full reproducibility of the work.

Ideally, the plasmids or their sequences should be deposited in a repository or attached as supplementary data to this manuscript.

pAM5823: no problems

pAM5825: the assembly of the pAM4950 linearised backbone with PconII RBS and cotA fragment cannot be performed with primers indicated in Table S3. The problem exists in the 5`end of the cotA fragment. Either one extra primer pair is used for the promoter-RBS fragment amplification, or the promoter-RBS fragment generation is missing. Moreover, the primer cotA_5051-F can be used to generate riboswitch plasmid but not a constitutive expression plasmid.

pAM5826: construction of this plasmid is based on the backbone of pAM5051 (Ma et al. 2014). The study does not contain sufficient information to replicate the construct. More information should be provided so that the work can be reproduced.

pAM5829: construction of this plasmid lacks sufficient detail to allow reproducibility. The primer pair 0766_riboF_pAM4940-F 0766_riboF_pAM4940-R lacks a template description showing what fragment was assembled with the backbone.

Apply recommendations throughout the constructs and either provide sequences of final plasmids or prepare the plasmid generation flowchart that clearly explains what fragments are used with a scheme of descendants that would allow replicating the work.

.

Transformation confirmation

The proposed primer pairs allowing for testing the successful transformation are sub-optimal.

Both primer pairs (NS1_Screen-F and NS1_Screen-R; NS2_Screen-F NS2_Screen-R) bind closely to the antibiotic resistance cassette and transgene integration site.

The amplification products of these primers may indicate the successful transformation, but they are by no means indicative of successful integration into the chromosome. The selected primer pairs will give the same result in the case of two events:

- a. Successful integration into the genome
- b. Extrachromosomal replication (both RSF1010 and Col1 pMA-RQ types of plasmids have been found to replicate independently of the chromosome in 7942)

The pair of primers, or at least one primer, external to the integration site in the plasmid should be used to verify the integration. Alternatively, the claims about successful chromosomal integration should be abolished and corresponding figures modified.

Localisation of cotA protein.

The previous work on the expression of laccases in 7942 was largely inconclusive concerning the location of the catalysis. Since the submitted manuscript uses a similar design i.e. the cotA protein lacks any secretion signal, it would be good if the authors showed where the catalysis takes place.

In principle, in the absence of a predesigned protein export, there are four possible scenarios:

1. Catalysis occurs intracellularly, the cotA remains in the cytoplasm or the membrane, and the substrate migrates in and, after catalysis, migrates out.
2. The catalysis takes place extracellularly via a mediator molecule that travels between the cytoplasm and the growth medium
3. Catalysis occurs through a stable cotA laccase that remained after the 7942 have lysed, and neither substrate nor mediator enters the cell

4. The laccase is exported to the growth medium with other known or unknown mechanisms, and catalysis occurs there.

Minor issues:

Figure 1: the structure of ethylbenzene with three leaves is irrelevant to the work.

The red dot (pollutant) input proposed in the scheme does not correspond to the theophylline riboswitch proposed in this work

Figure 2: larger FESEM and confocal images should be included in supplementary materials

Figure 2c: brightfield panel is missing

Figure S1 Table S1: these could be homogenised and combined. The honeycomb (7) and Grid_B lack photos. More optimal data presentation would include the table containing corresponding photographs of the printed patterns.

Figure 3: Lack of consistency between micrographs containing 3 strains (YFP+, YFP- RiboYFP+) and bar graph containing 4 strains (YFP+, YFP- RiboYFP+, WT), Scale bars are missing on the micrographs

Figure 4d: Please explain why those plates have a pink background.

Figure 4e: Description of *, **, ns is lacking in the figure legend

Figure 5c: The legend and marking should be clearer and indicate on the figure which of the datasets correspond to the ligand-containing data points and which not.

Figure 5b,d: Please explain why some of the photo plates have a pink background and others do not.

Figure S6: Description of *, **, ***, **** is lacking in the figure legend

P9 “decrease in biomass towards the center of the hydrogel, suggesting a limitation of gas and/or nutrient exchange (Figure 2a)”: fragment contradicts earlier section “The transparency of the hydrogel allows for light penetration, while the presence of relatively large pores within the matrix facilitates efficient diffusion of gas and nutrients”.

Please find a consensus between the two statements.

P16 Authors should elaborate on why Laccase--riboF-Lysis- and Laccase+-riboF-Lysis+ saturated with ABTS solution were placed on sterile agar LB plates for activity testing.

Was the growth of opportunistic bacteria described earlier on P11 observed?

P27 “The growth of WT *S. elongatus* cells was monitored and optimized in different geometries of 3D printed patterns over time. Specifically, cells encapsulated in three different printed structures (disk, honeycomb, and grid_A) were incubated in BG-11 medium with constant illumination for 7 days and were visually monitored for color density over time. Images were taken at regular time intervals. “

The method could be more quantitative, applying ImageJ or Metamorph, similar to YFP figures.

P30 Were the extracellular and pellet fractions analysed for the presence of CotA?

Reviewer #2 (Remarks to the Author):

I really enjoyed reading this paper. The authors built a solid foundation for creating photosynthetic ELMs and characterized key parameters critical for 3-D printing cyanobacteria-based bioink. The experiments are well designed, carefully executed, and clearly presented. I do feel the input, output, and abiotic materials used in this work could benefit from a few new elements. However, cyanobacteria engineering is time-consuming, and this work is already a great example of adding genetic programmability to cyanobacteria-based ELMs. I believe this paper is exciting for both the ELM community and the general readership of the journal. There are a few points that need to be addressed to improve the clarity of this paper before its publication.

1. “when supplemented with the small molecule theophylline, undergoes a conformational change allowing the mRNA to bind to an RBS, resulting in the translation of a target protein” reads a bit confusing. Shouldn’t the theophylline binding expose the RBS?
2. The mismatch between the target chemical species involved in the sensing and remediation modules might confuse the readers with limited background knowledge. In fact, Figure 1 is somewhat misleading by having the same visual representation for both the input and the pollutant. Also, the part about theophylline-induced laccase expression reads like a redundant detour, given that the laccase was constitutively expressed in the final strain, and the theophylline induction was already proven using YFP expression. The logical flow could be improved here.
3. Was calcium ion provided in the media during prolonged incubation, or were the ions in BG11 enough to prevent gel expansion/dissociation? Could the authors also comment on the availability of ions necessary for maintaining the structural integrity of alginate-based ELMs while deployed in real-world freshwater?
4. On Page 15, is there a reason why the incubation with ABTS took 24 hours?
5. On page 16, some extra information about the cell lysis mechanism would be helpful for readers. I understand there are more details on page 20, but feel it would be nice to have those at the term's first appearance.
6. In Figure 5c, the legends right next to the lines need to be non-repetitive and more specific. Please consider adding information about whether theophylline was added.

7. The current version of the RiboF glyph used by the authors is more commonly recognized for RBS only. Please consider adding an aptamer (<https://sbolstandard.org/visual-glyphs/>) before the RBS semicircle.

8. Were the integrations fully segregated? Does that change the laccase activity or lysis efficiency observed?

Reviewer #3 (Remarks to the Author):

The manuscript by Datta et al. designs a stimuli-responsive cyanobacterial-based ELM system, coupled with 3D printing technique, to achieve multiple functional outputs when exposed to a chemical stimulus. The authors fundamentally explored the 3D printed material properties, cell viability, and biological activities of the ELMs. Genetic toolkits are embedded in this ELM system to fabricate stimuli-responsive ELMs with functional outputs and tailored regulatory circuits. The idea and effort to solve the concern of the unintended biocontamination from the ELMs is exciting.

This study is solid and innovative for the field of Engineered Living Materials. I only have a few minor comments that the authors may wish to address.

1) In Figure 1, the schematic polymer network from zoom-in view is confusing. Traditional polymer networks are usually randomly entangled and crosslinked.

2) In Figure 2b, it is hard to tell whether these false green areas are individual cells/colonies of cells or not, given the average *S. elongatus* cell size is 2 μm .

3) Insufficient evidence is provided to prove that surface area to volume ratio is a vital factor affecting the growth behavior of the ELMs. The reason is because, in table S1, the length, thickness, number of layers, and printed sample volume for 3 patterns are all different. Further evidence needs to be shown to clearly demonstrate only the factor (surface area to volume ratio) will affect the growth.

4) In Figure 2c, are these images at day 0, 5, and 7 taken at the same spot of the sample?

5) How does the authors measure the number of incident irradiance with the unit ($\mu\text{mol photons m}^{-2} \text{s}^{-1}$)? Details need to be provided in the method section.

6) In Figure 4b and 4f, statistical analysis is recommended.

7) In Figure 4e, it is unclear why there is a significant difference between Laccase-riboF-Lysis and unloaded samples in the first two days.

8) Scale bars are missing in all the photos. Number of samples tested needs to be clearly identified in the figure captions.

Overall, I would recommend this study for publication with these minor suggested revisions.

Please see our point by point response to the reviewers' comments. Comments from the reviewers are in bold font, our response is in italics, and text in red indicates changes made to the text or additional/revised figures. Likewise, changes in the text of the final manuscript are also indicated in red with highlighting.

Reviewer(s)' Comments to Author:

REVIEWER COMMENTS

Reviewer #1 (Remarks to the Author):

Summary:

The manuscript describes the development of Engineered Living Material comprising an alginate scaffold containing photosynthetically active cyanobacteria capable of constitutive/induced laccase enzyme production and induced cell lysis. The proposed light and stimuli-responsive biomaterial is an exciting solution for the bio-contained bioremediation of textile dyes and could be explored further for even more interesting applications with more sophisticated synthetic biology circuits. Overall it is an exciting manuscript at the technical interface of synthetic biology and material science, focusing on the topics of biocatalysis and bioremediation. An improved manuscript will be attractive to people working in numerous, diverse fields. The conclusions of the manuscript are supported by the data.

The reference list could be improved. For example, it lacks foundation work on CotA laccase by L.O. Martins group, and on laccases more broadly.

We thank the reviewer for the suggestion and have updated the reference list to include more foundational work on CotA, have updated the text in the manuscript, and have included the following references:

- a) Martins, L. O., Soares, C. M., Pereira, M. M., Teixeira, M., Costa, T., Jones, G. H., & Henriques, A. O. (2002). Molecular and biochemical characterization of a highly stable bacterial laccase that occurs as a structural component of the *Bacillus subtilis* endospore coat. *Journal of Biological Chemistry*, 277(21), 18849-18859.
- b) Durao, P., Chen, Z., Fernandes, A. T., Hildebrandt, P., Murgida, D. H., Todorovic, S., ... & Martins, L. O. (2008). Copper incorporation into recombinant CotA laccase from *Bacillus subtilis*: characterization of fully copper loaded enzymes. *JBIC Journal of Biological Inorganic Chemistry*, 13, 183-193.
- c) Hullo, M. F., Moszer, I., Danchin, A., & Martin-Verstraete, I. (2001). CotA of *Bacillus subtilis* is a copper-dependent laccase. *Journal of bacteriology*, 183(18), 5426-5430.

*The following text has been added (page 14 and 15): “The CotA laccase from *Bacillus subtilis* was chosen, as extensive work characterizing the copper dependent enzyme has demonstrated its capacity for oxidizing synthetic dyes, and it has been previously expressed in cyanobacteria 35,42–44. “*

The manuscript is well written and, overall, clearly presents the data. However, the methodology section of the work requires significant improvement to meet the standard required by Nature Communications. Please see the following recommendations below:

Major issues:

The construction of plasmids

While the overall plan of plasmid construction is understandable and could be approximated by someone skilled in the art, the experimental description presented in the current version of the manuscript is insufficient to replicate the experiment fully. Therefore, the section should be corrected in the revised manuscript to allow full reproducibility of the work. Ideally, the plasmids or their sequences should be deposited in a repository or attached as supplementary data to this manuscript.

Thank you for your attention to detail. We have now included the plasmid sequences as supplementary data to the manuscript. We have also made edits to the text that we believe further clarify the construction of the plasmids and feel that all work is now fully reproducible. Please see below for specific changes that have been made.

pAM5823: no problems

pAM5825: the assembly of the pAM4950 linearised backbone with PconII RBS and cotA fragment cannot be performed with primers indicated in Table S3. The problem exists in the 5` end of the cotA fragment. Either one extra primer pair is used for the promoter-RBS fragment amplification, or the promoter-RBS fragment generation is missing. Moreover, the primer cotA_5051-F can be used to generate riboswitch plasmid but not a constitutive expression plasmid.

pAM5825 was not created using any primers. As mentioned in the methods section, a cotA gBlocks gene fragment was synthesized by Integrated DNA Technologies with 20 nucleotide overhangs with pAM4950 for Gibson assembly. However, we realize that we did not state that a PconII and RBS were included in the synthesized gene block upstream of the cotA sequence.

We have modified the text in the methods to clarify this point (page 33): “Plasmid pAM5825 was created by digesting pAM4950 with SwaI and performing a Gibson Assembly (NEB) between the linearized product and a cotA gBlocks Gene Fragment synthesized by Integrated DNA Technologies (IDT) containing a PconII promoter and RBS upstream of the codon optimized cotA sequence.”

Additionally, we have included the sequence of the fragment synthesized by IDT, as well as the pAM5825 plasmid map, in the updated SI. We included Supplementary Table S2 with the list of detailed sequence files used in this study and their respective description in the Supporting information.

pAM5826: construction of this plasmid is based on the backbone of pAM5051 (Ma et al. 2014). The study does not contain sufficient information to replicate the construct. More information should be provided so that the work can be reproduced.

With the addition of the plasmid maps for pAM5826, pAM5825 and pAM5051 in Supplementary Table S2, we feel that sufficient information is provided to replicate the construct. As noted in the methodology section:

“pAM5826 was created by digesting pAM5051 with EcoRI (NEB), and performing a Gibson Assembly between the linearized product and the PCR product of pAM5825 with primer pair cotA_5051-F/cotA_5051-R”.

pAM5829: construction of this plasmid lacks sufficient detail to allow reproducibility. The primer pair 0766_riboF_pAM4940-F 0766_riboF_pAM4940-R lacks a template description showing what fragment was assembled with the backbone.

Apply recommendations throughout the constructs and either provide sequences of final plasmids or prepare the plasmid generation flowchart that clearly explains what fragments are used with a scheme of descendants that would allow replicating the work.

We have updated Supplementary Table S2 in the SI to include the sequence of pAM5829. Additionally, we have updated the text (page 33) to describe the template target:

*“pAM5829 was created by digesting pAM4940 with SwaI and performing a Gibson Assembly of the linearized product and the PCR product of *S. elongatus* gDNA and the primer pair 0766_riboF_pAM4940-F/0766_riboF_pAM4940-R targeting the amplification of *Synpcc7942_0766*”.*

We believe that with the pAM5829 sequence provided and text updated, the gDNA template used should be clear and the construct reproducible.

Transformation confirmation

The proposed primer pairs allowing for testing the successful transformation are sub optimal. Both primer pairs (NS1_Screen-F and NS1_Screen-R; NS2_Screen-F NS2_Screen-R) bind closely to the antibiotic resistance cassette and transgene integration site. The amplification products of these primers may indicate the successful transformation, but they are by no means indicative of successful integration into the chromosome.

The selected primer pairs will give the same result in the case of two events:

- a. Successful integration into the genome**
- b. Extrachromosomal replication (both RSF1010 and Col1 pMA-RQ types of plasmids have been found to replicate independently of the chromosome in 7942)**

The pair of primers, or at least one primer, external to the integration site in the plasmid should be used to verify the integration. Alternatively, the claims about successful chromosomal integration should be abolished and corresponding figures modified.

Thank you for your concern regarding the validity of the chromosomal integration verification. Please note that the primer pairs (NS1_Screen-F and NS1_Screen-R; NS2_Screen-F NS2_Screen-R) were only used for chromosomal integration when transformations were conducted with suicide plasmids that will not replicate in the host.

For plasmids with extrachromosomal replication, (i.e. RSF1010) we did not target chromosomal integration, and these plasmids do not have arms for recombination. Please note that in this study, the YFP strains with RSF1010 backbones (YFP⁺, YFP⁻, and Ribo-YFP⁺) were not tested for chromosomal integration, nor were they expected to integrate into the chromosome.

All other strains used PCR to verify chromosome integration. This strategy of using primers with homology to the left and right arms of integration sites has been used extensively in the Golden cyanobacterial genetics lab for checking successful integration into the chromosome for decades.

*While primers outside of integration arms may be ideal, we are using suicide plasmids that are unable to replicate in *S. elongatus*. As the cells are grown on selective media, typically, cells transformed with a plasmid with arms for recombination but without the capacity for extrachromosomal replication will not survive on the selective agar/media. We use ~800 nucleotides of homologous sequence on each side of the integration site limiting the risks of off-target integration. Additionally, DNA transfer through natural selection favors double recombination over single recombination. We assess double recombination as the presence of only one larger band containing the insert after PCR, as opposed to two bands (with the smaller band containing the unmodified WT region). While PCR may be prone to false positive results, with the use of proper controls (mutant, blank, and wildtype), PCR is a valid tool and widely used by many other labs as well. This strategy is routinely successful in *Synechococcus* PCC 7942.*

The text has been updated to clarify that chromosomal integration was not expected during transformation of the YFP strains (page 34):

*“Plasmids pAM4909, pAM5027, and pAM5057 were transferred into *S. elongatus* by biparental conjugation on selective media⁵⁴. Transformation of *S. elongatus* with all other plasmids was achieved using standard protocols⁵⁵. All *S. elongatus* transformations were performed using WT strain AMC06. Genotyping of *S. elongatus* strains expected to contain integration into a chromosome neutral site (not including YFP⁺, YFP⁻, and Ribo-YFP⁺ which bear self-replicating extrachromosomal plasmids) was*

performed using colony PCR with Q5 DNA Polymerase (NEB), using primer pair NS1_Screen-F/NS1_Screen-R for transformations in neutral site 1 (NS1), and NS2_Screen-F/NS2_Screen-R for transformation in neutral site 2 (NS2). A complete list of strains used in this study is included in Table S4.”

Localisation of cotA protein.

The previous work on the expression of laccases in 7942 was largely inconclusive concerning the location of the catalysis. Since the submitted manuscript uses a similar design i.e. the cotA protein lacks any secretion signal, it would be good if the authors showed where the catalysis takes place.

In principle, in the absence of a predesigned protein export, there are four possible scenarios:

- 1. Catalysis occurs intracellularly, the cotA remains in the cytoplasm or the membrane, and the substrate migrates in and, after catalysis, migrates out.**
- 2. The catalysis takes place extracellularly via a mediator molecule that travels between the cytoplasm and the growth medium**
- 3. Catalysis occurs through a stable cotA laccase that remained after the 7942 have lysed, and neither substrate nor mediator enters the cell**
- 4. The laccase is exported to the growth medium with other known or unknown mechanisms, and catalysis occurs there.**

The authors agree with the reviewer that this information would be helpful and is an excellent suggestion. The authors have tried to answer the question of location of catalysis but have not been able to demonstrate conclusive proof, thus far, due to the complexity of the system. Analyses of the cellular pellet and supernatant fractions of liquid cultures indicate that active laccase is not present in the supernatant during initial growth in BG-11 liquid media, suggesting that catalysis either occurs intracellularly or extracellularly following cell lysis and release of a stable cotA protein. We are currently investigating the expression of cotA proteins with different secretion signals. Our experiments, thus far, remain inconclusive but if successful this would open a plethora of opportunities if the protein activity and export mechanism could be tracked. Given an absence of a predesigned protein export mechanism for the current study, the authors have included the possible scenarios (page 23) as suggested and thank the reviewer for the excellent feedback:

“Given the complexity of the composite material system, we were unsuccessful in identifying the localization of the CotA protein in the ELM. There are several potential scenarios under which catalysis could occur, including intracellular catalysis in which the substrate and/or mediator molecules cross into the cell and the product is exported or diffuses out of the cell, extracellular catalysis in which the laccase is exported from the cell, extracellular catalysis by a stable laccase following cell lysis, or likely a combination of the above.”

Minor issues:

Figure 1: the structure of ethylbenzene with three leaves is irrelevant to the work. The red dot (pollutant) input proposed in the scheme does not correspond to the theophylline riboswitch proposed in this work

As per the reviewer's suggestion the structure has been deleted from Figure 1. We have further edited the figure by altering the colors of the dots representing the input, biosynthetic output, pollutant, and inert pollutant, considering suggestions from both reviewers' 1 and 2.

Figure 2: larger FESEM and confocal images should be included in supplementary materials

As suggested by the reviewer we have included larger FESEM images in the supplementary information (Supplementary Figure S2). Additionally, we include larger confocal images in the supplementary information (Supplementary Figure S3-4).

Figure 2c: brightfield panel is missing

We captured and included the brightfield panel for day 7 samples. Additional confocal images with the brightfield panel have been added to the supplementary information (Supplementary Figure S5).

Figure S1 Table S1: these could be homogenised and combined. The honeycomb (7) and Grid_B lack photos. More optimal data presentation would include the table containing corresponding photographs of the printed patterns.

As suggested by the reviewer, we have combined Supplementary Figure S1 and Table S1 into a table containing photographs of the printed patterns. We have updated Supplementary Table S1 along with the honeycomb (7) and Grid_B photos.

Figure 3: Lack of consistency between micrographs containing 3 strains (YFP+, YFP and RiboYFP+) and bar graph containing 4 strains (YFP+, YFP- RiboYFP+, WT), Scale bars are missing on the micrographs

We have added the scale bars on the missing micrographs (Figure 3b). The WT data has been added and is available as supplementary information (Supplementary Figure S10).

Figure 4d: Please explain why those plates have a pink background.

Figure 4d in the previous version of the manuscript is now numbered as figure 4b in the revised version.
The authors apologize for the difference. While taking photographs of experiments, some of the samples were placed on the lab work bench and they do not have pink backgrounds. Later samples were placed above a light illumination setup (see photograph below), for better image clarity, but the photographs obtained a pinkish background due to the light setup that was used during compilation of the images.

Photograph Setup 1
On lab workbench

Photograph in Light Setup 2
On lab workbench

Photographs taken in Setup 1

Photographs taken in Setup 2

Figure 4e: Description of *, **, ns is lacking in the figure legend

Figure 4e in the previous version of the manuscript is now numbered as figure 4c in the revised version.

The figure legend has been updated to include a description of the asterisks now used through figure 4:

“, **, ***, and ns indicate P values ≤ 0.05 , 0.01, 0.001, and not significant, respectively.”*

Figure 5c: The legend and marking should be clearer and indicate on the figure which of the datasets correspond to the ligand-containing data points and which not.

The graph has been updated with an appropriate legend, as suggested.

Figure 5b,d: Please explain why some of the photo plates have a pink background and others do not.

As mentioned above, this discrepancy is due to variation in the lighting arrangement that the photographs were taken under in the laboratory.

Figure S6: Description of *, **, *, **** is lacking in the figure legend**

Figure S6 in the previous version of the manuscript is now numbered as figure S14 in the revised version. The figure legend has been updated to describe the meaning of the asterisks:

*“Laccase activity of the supernatant of media surrounding hydrogels. Time-course of the oxidation of ABTS in reaction buffer with the addition of supernatant surrounding either the Laccase⁻-riboF-Lysis⁻, Laccase⁺-riboF-Lysis⁺ strain, or an unloaded hydrogel. *, **, ***, and **** indicate P values ≤ 0.05 , 0.01, 0.001 and 0.0001, respectively.”*

P9 “decrease in biomass towards the center of the hydrogel, suggesting a limitation of gas and/or nutrient exchange (Figure 2a)”: fragment contradicts earlier section “The transparency of the hydrogel allows for light penetration, while the presence of relatively large pores within the matrix facilitates efficient diffusion of gas and nutrients”. Please find a consensus between the two statements.

We have edited the second sentence by removing the word “efficient” (page 7). We feel this conveys that media and gas can diffuse at some level, as supported by the reference, but doesn’t suggest it is infallible at scale:

“The transparency of the hydrogel allows for light penetration, while the presence of relatively large pores within the matrix facilitates diffusion of gas and nutrients.³⁹”

P16 Authors should elaborate on why Laccase--riboF-Lysis- and Laccase+-riboF-Lysis+ saturated with ABTS solution were placed on sterile agar LB plates for activity testing. Was the growth of opportunistic bacteria described earlier on P11 observed?

We wanted to evaluate whether any product leached onto/into the underlying agar, however, this was not apparent. We were also interested to see whether opportunistic bacteria would grow on the agar plate in contact with the hydrogel; however, bacterial growth was not seen. Ultimately, the samples were appropriate to use for photographic representation of the ABTS reaction within the hydrogel.

P27 “The growth of WT *S. elongatus* cells was monitored and optimized in different geometries of 3D printed patterns over time. Specifically, cells encapsulated in three different printed structures (disk, honeycomb, and grid_A) were incubated in BG-11 medium with constant illumination for 7

days and were visually monitored for color density over time. Images were taken at regular time intervals. “

The method could be more quantitative, applying ImageJ or Metamorph, similar to YFP Figures.

We appreciate this comment and have considered a more quantitative approach. We encountered significant hurdles when attempting to quantify growth. For example, we observed that cell growth is somewhat patchy and not uniform throughout the hydrogel across axes (both laterally and at depth), and thus it was difficult to count the number of cells or colonies in a 3D space, as opposed to a 2D structure, such as the surface of a petri dish. We observed a change in cell number per focal plane when a cross section of the cell containing hydrogel was imaged using z stacking in confocal microscopy. We have provided a representative 3D reconstructed image and video in support of this (Supplementary video S1, and Supplementary Figure S7).

We have also calculated the cell viability (percentage of live & dead cells) using Fiji software for sample images from the confocal data set using raw data. The data is provided in Supplementary Figure S6. While Figure 2c submitted in the original version of the manuscript had brightness enhanced uniformly across all the panels during postprocessing for better visualization of the SYTOX Blue channel, the quantitative analysis was conducted with raw data and this has been added in the methods section (Supplementary Figure S6). We have added the enlarged version of both in the SI (Supplementary Figure S3 and S4).

P30 Were the extracellular and pellet fractions analysed for the presence of CotA?

We analyzed the extracellular and pellet fractions for the presence of CotA during our preliminary optimization studies in liquid culture.

Methods: Cells were inoculated at OD₇₅₀ of 0.03 and incubated for 6 days to an OD₇₅₀ of approximately 1.0. After 6 days the samples were centrifuged at 4500 rcf for 10 min and the supernatant was separated from the pellet. Pellet fractions were sonicated on ice for 1 min and centrifuged at 4500 rcf for 10 min. The samples were then transferred to a 1 mL reaction buffer mixture of 100 mM sodium acetate buffer, 3.16 μM CuSO₄, pH 5.0, and 2 mM ABTS in a 1.5 ml conical microtubes and incubated in darkness at 28 °C. The oxidation of ABTS was determined by measuring the absorbance of the reaction mixture at 420 nm after 24 hours. A blank reading at 420 nm of the reaction mixture buffer was subtracted from all samples, and the signal was normalized to OD₇₅₀.

We did not observe any significant signal in the supernatant. There was a very low signal observed in the extracellular fraction of the laccase mutant versus no signal in the wildtype, but it was variable and typically about ~5-10% of the lysate fraction. We also observed a ~10-20% signal in supernatant during tests with higher concentrations of CuSO₄, although this was likely due to cell death/lysis from the toxicity of copper.

Reviewer #2 (Remarks to the Author):

I really enjoyed reading this paper. The authors built a solid foundation for creating photosynthetic ELMs and characterized key parameters critical for 3-D printing cyanobacteria-based bioink. The experiments are well designed, carefully executed, and clearly presented. I do feel the input, output, and abiotic materials used in this work could benefit from a few new elements. However, cyanobacteria engineering is time-consuming, and this work is already a great example of adding genetic programmability to cyanobacteria-based ELMs. I believe this paper is exciting for both the

ELM community and the general readership of the journal. There are a few points that need to be addressed to improve the clarity of this paper before its publication.

1. “when supplemented with the small molecule theophylline, undergoes a conformational change allowing the mRNA to bind to an RBS, resulting in the translation of a target protein” reads a bit confusing. Shouldn’t the theophylline binding expose the RBS?

We agree that this sentence was confusing. We have reworded it to improve clarity (page 4):

*“Previous studies have demonstrated the utility of the theophylline-responsive riboswitch-F in the cyanobacterium *Synechococcus elongatus* PCC 7942 (*S. elongatus*), which, when supplemented with the small molecule theophylline, undergoes a conformational change exposing the RBS and allows the mRNA to bind to a ribosomal subunit, resulting in the translation of a target protein³².”*

2. The mismatch between the target chemical species involved in the sensing and remediation modules might confuse the readers with limited background knowledge. In fact, Figure 1 is somewhat misleading by having the same visual representation for both the input and the pollutant. Also, the part about theophylline-induced laccase expression reads like a redundant detour, given that the laccase was constitutively expressed in the final strain, and the theophylline induction was already proven using YFP expression. The logical flow could be improved here.

*We have edited figure 1 using different colors for the input and the pollutant. We have moved *riboF-cotA* data to SI (Supplementary Figure S11 and Supplementary text “Construction of riboswitch-induced laccase strains and ABTS-Laccase activity testing for 3D printed constructs”) for better logical flow of the manuscript.*

3. Was calcium ion provided in the media during prolonged incubation, or were the ions in BG11 enough to prevent gel expansion/dissociation? Could the authors also comment on the availability of ions necessary for maintaining the structural integrity of alginate-based ELMs while deployed in real-world freshwater?

The additional calcium ions for crosslinking were only provided during fabrication steps. During prolonged incubation, the hydrogels were submerged solely in BG-11 media. However, we did not probe the effect of the ions in gel expansion or dissociation. Visually, the gel structures were intact even after prolonged incubation of at least 12 days under our experimental conditions in BG-11, and we have observed that the hydrogel is stable in DI water for at least 2 years. As BG-11 is a complete medium designed for the isolation of cyanobacteria, we expect the gel to retain structural integrity in real-world freshwater conditions.

4. On Page 15, is there a reason why the incubation with ABTS took 24 hours?

We found that the signal was much stronger and less noisy after 24 hours relative to shorter reaction times (i.e., 60 seconds to 1 hour) during the optimizing of our experiments. This may be due, in part, to the rate of diffusion of enzymes and substrates into and out of the hydrogel matrix.

5. On page 16, some extra information about the cell lysis mechanism would be helpful for readers. I understand there are more details on page 20, but feel it would be nice to have those at the term's first appearance.

We agree that it would be appropriate to include a mention of the cell lysis mechanisms at this point in the paper. We have included the following sentence on page 15:

*“The overexpression of the *Synpcc7942_0766* gene results in the excision of a prophage from the *S. elongatus* genome, causing cellular lysis.”*

6. In Figure 5c, the legends right next to the lines need to be non-repetitive and more specific. Please consider adding information about whether theophylline was added.

We have corrected and replaced the graph in Figure 5c. The theophylline addition has been mentioned in both the figure legend and the figure description below.

7. The current version of the RiboF glyph used by the authors is more commonly recognized for RBS only. Please consider adding an aptamer (<https://sbolstandard.org/visual-glyphs/>) before the RBS semicircle.

Thank you for the suggestion. The aptamer has been added to Figure 1, and the genetic circuits in Figures 3, 4 and Supplementary Figure S11a.

8. Were the integrations fully segregated? Does that change the laccase activity or lysis efficiency observed?

As evidenced by PCR amplification of sites of integration, the strains with laccase/lysis capabilities were fully segregated. It is conceivable that fully segregated strains would have higher activity than partial.

Reviewer #3 (Remarks to the Author):

The manuscript by Datta et al. designs a stimuli-responsive cyanobacterial-based ELM system, coupled with 3D printing technique, to achieve multiple functional outputs when exposed to a chemical stimulus. The authors fundamentally explored the 3D printed material properties, cell viability, and biological activities of the ELMs. Genetic toolkits are embedded in this ELM system to fabricate stimuli-responsive ELMs with functional outputs and tailored regulatory circuits. The idea and effort to solve the concern of the unintended biocontamination from the ELMs is exciting.

This study is solid and innovative for the field of Engineered Living Materials. I only have a few minor comments that the authors may wish to address.

1) In Figure 1, the schematic polymer network from zoom-in view is confusing. Traditional polymer networks are usually randomly entangled and crosslinked.

Thank you, Figure 1 has been corrected.

2) In Figure 2b, it is hard to tell whether these false green areas are individual cells/colonies of cells or not, given the average *S. elongatus* cell size is 2 μm .

Larger FESEM images have been added to the supplementary to provide better visualization of the cells (Supplementary Figure S2). Additional FESEM micrographs given in the Supplementary Information show both individual cells and cell colonies (Supplementary Figure S2 c-e).

3) Insufficient evidence is provided to prove that surface area to volume ratio is a vital factor affecting the growth behavior of the ELMs. The reason is because, in table S1, the length, thickness, number of layers, and printed sample volume for 3 patterns are all different. Further evidence needs to be shown to clearly demonstrate only the factor (surface area to volume ratio) will affect the growth.

Thank you for this comment. We have updated this section to include the following sentence (page 9):

“While an enhanced surface area to volume ratio was targeted in the study, the thickness as well as density of printed layers will also likely affect the growth behavior of cyanobacterial ELMs.”

4) In Figure 2c, are these images at day 0, 5, and 7 taken at the same spot of the sample?

*The hydrogel (Grid_A) was cut at different time points at random positions and confocal images were taken. Once the image was taken the cut piece was discarded to avoid contamination from the imaging process. We have updated the text in the methods (page 26 and 27) for “Cell viability of 3D-printed WT *S. elongatus*” to clarify this point:*

“Samples from different time points were imaged to monitor cell viability. Independent regions of the hydrogel were used for imaging of different timepoints to minimize contamination.”

5) How does the authors measure the number of incident irradiance with the unit ($\mu\text{mol photons m}^{-2} \text{s}^{-1}$)? Details need to be provided in the method section.

We thank the reviewer for bringing this to our attention. The following text has been added to the methods section (page 24):

“Irradiance measurements were conducted with a 4π quantum light meter (model QSL-100, Biospherical Instruments)”

0) In Figure 4b and 4f, statistical analysis is recommended.

Figure 4b and 4f in the previous version of the manuscript are now numbered as figure Suub and figure 4d respectively in the revised version.

The authors thank the review for the suggestion. We have added the statistical analysis for these figures.

1) In Figure 4e, it is unclear why there is a significant difference between Laccase-riboF-Lysis and unloaded samples in the first two days.

Figure 4e in the previous version of the manuscript is now numbered as figure 4c in the revised version.

There appear to be significant differences in oxidative activity between blank hydrogels and those containing wild-type cells (or relevant controls). This is likely due to oxidative enzymes native to the strain that exhibit some level of activity against ABTS (and other substrates).

2) Scale bars are missing in all the photos. Number of samples tested needs to be clearly identified in the figure captions.

We apologize for the oversight and have added scale bars in the images and provided the number of samples in the figure captions. We have also updated the number of samples tested in all figure captions.

Overall, I would recommend this study for publication with these minor suggested revisions.

REVIEWER COMMENTS

Reviewer #1 (Remarks to the Author):

The manuscript of Datta et al has been markedly improved in the first round of revisions and mostly meets the criteria for publication in Nature Communications.

Despite those improvements, there are still three areas where additional revisions should be implemented.

1. Transformation and integration of the transgenes into *Synechococcus* 7942 chromosome.

The presented results do not confirm the full segregation of the transgenic strains.

When it comes to this application the full segregation of CotA expression is not essential and even non-homogenous cells will yield a functional biomaterial.

Authors claim that the used plasmids are suicidal (presumably meaning unable to replicate in 7942). As such, every transformant that shows desired phenotypic feature and has a positive PCR result is considered fully segregated.

This is only partially correct, in other labs, NS 1 NS2 targeting vectors containing ColE1/pMB1 /pBR322/pUC oris have been found to replicate in 7942, persist for generations, and full sequences of plasmids are observed in routine resequencing of transgenic strains and control PCRs done with ori specific primers.

Again, for this application in ELM full segregation is not essential so if authors are less categorical about full segregation and restrict statements to successful transformation and specific integration, I would be happy with that. Alternatively, if the statements on full segregation are categorical, I would expect more proofs such as a combination of external to the vector and ori-specific pcrs and/or strain resequencing.

2. The larger micrographs have been now presented in the SI. This is a positive step forward for clarity of data presentation, another step would be to replace them with micrographs that are not only enlarged but also of higher resolution so that the interpretation of the data is easier. This is especially problematic for figures S9 and S10. Generally speaking, pdfs are significantly lower quality than original docx files and the latter should be exported at a higher resolution at the very least and replaced with better-resolution photos at the most.

3. The final construct of the project should be reconfirmed with PCR or WGS. Current confirmation does not meet these criteria as it focuses exclusively on CotA expression (Fig S12).

In Figure S12 the two parallel CotA constructs are tested NS1 Lac+ riboF Lysis+ at NS1 site and NS2 Lac+ riboF Lysis+ at NS2 site. Meanwhile, the manuscript exclusively focuses on NS2 Lac+ riboF Lysis+ site.

Unfortunately, there are no confirmatory PCRs for the introduction of the riboswitch at the 0766 locus of the genome.

Furthermore, assuming that the cassette of both constructs was identical there is a discrepancy in the size of PCR products. The NS2 construct looks considerably larger than NS1 (Fig S12). Considering that the empty NS2 site PCR product should be smaller than that of NS1, so should be the trend for the two sites with the same insert. Meanwhile, Figures S12 and S12a do not reflect this situation.

The section on strain construction and validation should be clarified in the revised manuscript regardless of the segregation of these strains.

Reviewer #2 (Remarks to the Author):

The authors addressed most of the comments nicely. I would love to congratulate them for doing a great job in revising. My only additional suggestion is to include a brief discussion about the potential swelling/gel expansion in different aquatic environments. Ionically crosslinked alginate hydrogels could lose >60% of their initial mechanical strength within 15 hours due to ion exchange with monovalent ions in the surrounding. Personally, I have witnessed obvious expansion and weakening of Ca-alginate hydrogels (encapsulating living bacteria) in environmental freshwater samples. With no quantitative data, it is hard to imagine how the ELMs presented here could maintain long-term structural integrity without a constant, sufficient supply of divalent ions. I understand this is not the paper's focus, but I find it helpful to mention it since biocontainment is a feature.

Reviewer #3 (Remarks to the Author):

The authors have addressed all my questions nicely. This work is exciting in the ELMs field, and I recommend this study for publication.

Please see our point by point response to the reviewers' comments. Comments from the reviewers are in bold font, our response is in italics, and text in red indicates changes made to the text or additional/revised figures. Likewise, changes in the text of the final manuscript are also indicated in red with highlighting.

Reviewer(s)' Comments to Author:

REVIEWER COMMENTS

Reviewer #1 (Remarks to the Author):

The manuscript of Datta et al has been markedly improved in the first round of revisions and mostly meets the criteria for publication in Nature Communications. Despite those improvements, there are still three areas where additional revisions should be implemented.

1. Transformation and integration of the transgenes into *Synechococcus* 7942 chromosome. The presented results do not confirm the full segregation of the transgenic strains. When it comes to this application the full segregation of CotA expression is not essential and even non-homogenous cells will yield a functional biomaterial. Authors claim that the used plasmids are suicidal (presumably meaning unable to replicate in 7942). As such, every transformant that shows desired phenotypic feature and has a positive PCR result is considered fully segregated. This is only partially correct, in other labs, NS1 NS2 targeting vectors containing ColE1/pMB1 /pBR322/pUC oris have been found to replicate in 7942, persist for generations, and full sequences of plasmids are observed in routine resequencing of transgenic strains and control PCRs done with ori specific primers.

Again, for this application in ELM full segregation is not essential so if authors are less categorical about full segregation and restrict statements to successful transformation and specific integration, I would be happy with that. Alternatively, if the statements on full segregation are categorical, I would expect more proofs such as a combination of external to the vector and ori-specific pcrs and/or strain resequencing.

*We agree with the reviewer that a claim of complete segregation requires amplification from outside of the integration point. We have opted to repeat the PCR proofs using primer pairs external to the arms of homology, thus the presence of bands of expected size is not a product of residual plasmid. **Supplementary table S3** primers NS1_Screen_F, NS1_Screen_R, NS2_Screen_F, and NS2_Screen_R have been updated accordingly with the new primer sequences, and **supplementary figure S12a has been updated** with a gel of the PCR products obtained using these new primer pairs for genotyping.*

Using these primer pairs, the expected band size for WT NS1 is 2013bp, Laccase⁺-riboF-Lysis⁺ NS1 is 3900 bp, RiboF-Laccase⁺ NS1 is 5029 bp, WT NS2 is 1539 bp, and Laccase⁺-riboF-Lysis⁺ NS2 is 4587 bp. Given that the PCR products reflect the expected bands, we wish to retain the statement of full segregation.

2. The larger micrographs have been now presented in the SI. This is a positive step forward for clarity of data presentation, another step would be to replace them with micrographs that are not only enlarged but also of higher resolution so that the interpretation of the data is easier. This is especially problematic for figures S9 and S10. Generally speaking, pdfs are significantly lower quality than original docx files and the latter should be exported at a higher resolution at the very least and replaced with better-resolution photos at the most.

*We thank the reviewers for bringing this to our attention. We have updated the images for **figures S9a-c and S10**. Additionally, we have now exported the original docx file at a higher resolution by changing the export settings during the docx to pdf conversion.*

3. The final construct of the project should be reconfirmed with PCR or WGS. Current confirmation does not meet these criteria as it focuses exclusively on CotA expression (Fig S12). In Figure S12 the two parallel CotA constructs are tested NS1 Lac+ riboF Lysis+ at NS1 site and NS2 Lac+ riboF Lysis+ at NS2 site. Meanwhile, the manuscript exclusively focuses on NS2 Lac+ riboF Lysis+ site. Unfortunately, there are no confirmatory PCRs for the introduction of the riboswitch at the 0766 locus of the genome.

Furthermore, assuming that the cassette of both constructs was identical there is a discrepancy in the size of PCR products. The NS2 construct looks considerably larger than NS1 (Fig S12). Considering that the empty NS2 site PCR product should be smaller than that of NS1, so should be the trend for the two sites with the same insert. Meanwhile, Figures S12 and S12a do not reflect this situation.

The section on strain construction and validation should be clarified in the revised manuscript regardless of the segregation of these strains.

We apologize for the confusion regarding the construction of Laccase⁺-riboF-Lysis⁺. To clarify, the riboswitch was not inserted before the 0766 locus of the genome; rather, plasmid AM5829 contains a copy of 0766 preceded by the riboswitch, with arms for integration at NS1. In the Laccase⁺-riboF-Lysis⁺ strain, the laccase⁺ cassette is inserted into the genome at the NS2 site (using AM5825) while the riboF-Lysis⁺ cassette is inserted at the NS1 site (using AM5829). This explains the discrepancy in the size of PCR products that is observed in S12a. Note that in this revision we have updated figure S12a for genotyping using primers suitable for verifying complete segregation at the NS1 and NS2 sites for the Laccase⁺-riboF-Lysis⁺ strain, and at the NS1 site for RiboF-Laccase⁺.

To clarify the construction of the Laccase⁺-riboF-Lysis⁺ strain in the text, we have updated the section on strain construction by correcting the following sentence (page 15) of the main manuscript:

“In addition to pAM5825, a neutral-site 1 (NS1) chromosome-integration plasmid (pAM5829; **Supplementary Table S2) containing a *conII* promoter–theophylline-responsive riboswitch upstream of a copy of the *S. elongatus* gene *Synpcc7942_0766* was constructed and inserted into the *S. elongatus***

chromosome at NS1 creating strain Laccase⁺-riboF-Lysis⁺, and printed in hydrogels in tandem with a corresponding control strain Laccase⁻-riboF-Lysis⁻.”

Reviewer #2 (Remarks to the Author):

The authors addressed most of the comments nicely. I would love to congratulate them for doing a great job in revising. My only additional suggestion is to include a brief discussion about the potential swelling/gel expansion in different aquatic environments. Ionically crosslinked alginate hydrogels could lose >60% of their initial mechanical strength within 15 hours due to ion exchange with monovalent ions in the surrounding. Personally, I have witnessed obvious expansion and weakening of Ca-alginate hydrogels (encapsulating living bacteria) in environmental freshwater samples. With no quantitative data, it is hard to imagine how the ELMs presented here could maintain long-term structural integrity without a constant, sufficient supply of divalent ions. I understand this is not the paper's focus, but I find it helpful to mention it since biocontainment is a feature.

We thank the reviewer for this excellent suggestion. We have now included the following section in the discussion (page 23) along with a reference to explain the effect of the ion concentration on the structural integrity of the alginate hydrogel.

“Structurally, factors such as the ratio of the β -D-mannuronic (M) and α -L-guluronic (G) acid (M/G) in the alginate, type of crosslinking (eg. chemical or ionic), and/or the concentration of divalent ions significantly affects the structural integrity and the degree of swelling of the hydrogel network. The presence of the divalent ions in the surrounding environment plays a crucial role in the crosslinking process and may result in weakening of the gel if Ca²⁺ ions are substituted^{52,53}.”

The following references have been added with the above text:

- a) Bialik-Wąs, K., Królicka, E., & Malina, D. (2021). Impact of the type of crosslinking agents on the properties of modified sodium alginate/poly (vinyl alcohol) hydrogels. *Molecules*, 26(8), 2381.
- b) Costa, M. J., Marques, A. M., Pastrana, L. M., Teixeira, J. A., Sillankorva, S. M., & Cerqueira, M. A. (2018). Physicochemical properties of alginate-based films: Effect of ionic crosslinking and mannuronic and guluronic acid ratio. *Food hydrocolloids*, 81, 442-448.

Reviewer #3 (Remarks to the Author):

The authors have addressed all my questions nicely. This work is exciting in the ELMs field, and I recommend this study for publication.

Thank you for the positive assessment of our work!